# Real-Time Estimation of Airflow Vector based on Lidar Observations for Preview Control

Ryota Kikuchi[1,2], Takashi Misaka[3], Shigeru Obayashi[4], Hamaki Inokuchi[2]

[1]DoerResearch Inc., Chiba 260-0013, Japan
[2]Japan Aerospace Exploration Agency, Tokyo 181-0015, Japan
[3]National Institute of Advanced Industrial Science and Technology, Ibaraki 305-8564, Japan
[4]Tohoku University, Miyagi 980-8577, Japan

*Correspondence to*: Ryota Kikuchi (Email: kikuchi-ryota@doerresearch.com)

**Abstract.** As part of control techniques, gust-alleviation systems using airborne Doppler Lidar technology are expected to enhance aviation safety by significantly reducing the risk of turbulence-related accidents. Accurate measurement and estimation of the vertical wind velocity are very important in the successful implementation of such systems. An estimation algorithm for the airflow vector based on data from airborne Lidars is proposed and investigated for preview control to prevent turbulence-induced aircraft accidents in flight. An existing technique — simple vector conversion— assumes that the wind field between the Lidars is homogeneous, but this assumption fails when turbulence occurs due to a large wind-velocity fluctuation. The proposed algorithm stores the line-of-sight (LOS) wind data at every moment and uses recent and past LOS wind data to estimate the airflow vector and to extrapolate the wind field between the airborne twin Lidars without the assumption of homogeneity. Two numerical experiments—using the ideal vortex model and numerical weather prediction, respectively—were conducted to evaluate the estimation performance of the proposed method. The proposed method has much better performance than simple vector conversion in both experiments, and it can estimate accurate two-dimensional wind-field distributions, unlike simple vector conversion. The estimation performance and the computational cost of the proposed method can satisfy the performance demand for preview control.

## 1 Introduction

Atmospheric turbulence poses a potential risk to aircraft operation. Statistics reported by Boeing (2018) show that 322 non-fatal and 51 fatal accidents occurred worldwide in commercial jet flights from 2009 through 2018. Of the fatal accidents, the largest proportion (25.5%) were due to Loss of Control-In Flight (LOC-I). The International Air Transportation Association (2016) shows that LOC-I frequently occurs when the aircraft speed is well below the stall speed; in conjunction with weather conditions, low speed is the most common factor in LOC-I accidents. Forty-two percent of LOC-I accidents occurred under degraded meteorological conditions affecting aircraft speed, in particular strong wind shear and atmospheric turbulence.

For both fatal and non-fatal aircraft accidents, the impact of atmospheric turbulence can be significant. The Japan Transport Safety Board has stated that accidents caused by turbulence accounted for 48% of non-fatal aircraft accidents in Japan involving commercial airplanes from 2003-2012. An increase in the rate of accidents related to turbulence was reported by the Federal Aviation Administration in 2006, Kim and Chun in 2011, and Williams in 2017. Accidents caused by convective systems such as cumulonimbus clouds have decreased owing to advances in airborne Radar (Airbus, 2020; Sermi et al. 2015). However, non-cloud atmospheric turbulence, called clear-air turbulence (CAT), cannot be detected by Radar, as reported by Soreide et al., 2000; Barny, 2012; and Inokuchi et al., 2009. Airborne CAT-observation systems to minimize risks of turbulence-related accidents are essential for aviation safety.

Numerical weather prediction (NWP), which is an essential tool for aircraft operation, can forecast weather conditions for days and even weeks in advance and output broader-area weather information than can Radar or Lidar. However, NWP cannot explicitly resolve disturbances as small as most turbulence, leading to a very large predictive uncertainty (Sharman et al. 2006, Kim et al. 2011). Therefore, some researchers have developed an alternative approach that predicts turbulence potential by calculating turbulence indicators from NWP results; for example, Sharman et al. (2006) have developed an approach called graphical turbulence guidance (GTG) that combines such indicators. The turbulence potential can also be used to determine operational flight routes (Kim et al. 2015), but it has a large spatio-temporal gap on the scale of aircraft motion because it is based on NWP results such as the meso-scale model. It thus provides insufficient information to implement turbulence avoidance on aircraft in flight.

Recently, airborne Doppler Lidar has been developed by Soreide et al., 2000; Barny, 2012; Inokuchi et al., 2009; Machida, 2017; and Inokuchi and Akiyama, 2019. Emitted laser light is scattered by fine aerosol particles in the atmosphere; the back-scattered light is condensed by telescopes and received by an optical transceiver. Since the wavelength of the received light varies according to the velocity of the aerosol particles due to the Doppler effect, wind speed can be calculated by comparing this wavelength with that of the received light (Inokuchi and Akiyama, 2019). However, when rain is too heavy, the backscattering signal is weakened due to strong attenuation by raindrops and a decrease in aerosols (Wei et. al 2019), making it difficult to measure the wind velocity at a distance. Japan Aerospace Exploration Agency (JAXA) is researching and developing a coherent Doppler Lidar capable of remotely detecting air turbulence in clear-air conditions, and has conducted a flight demonstration of a Lidar system that can provide turbulence information to pilots (Inokuchi et al., 2009; Machida, 2017; Inokuchi and Akiyama, 2019). Inokuchi et al. (2012) have shown observationally that airborne Doppler Lidar can detect CAT in front of an aircraft in flight at altitudes of 3,200 m; the Lidar information can be detected 30 seconds before the turbulence affects the aircraft. The aircraft's flight speed in the test was 320 kt (160 m/s), so it detected CAT from a distance of about 4.8 km.

Based on advance airflow information, flight demonstrations have been carried out with the aim of providing pilots with the information they need to make decisions: whether to change course to avoid wind shear, and whether to turn on seatbelt-sign lighting during cruise and altitude changes (Inokuchi and Akiyama, 2019). Although Lidar systems

are useful for providing onboard wind information to pilots, avoiding turbulence at high altitudes is difficult as the
range of detection that facilitates pilots to be warned is short (Hamada, 2019). Gathering such information involves
emitting a laser beam and receiving the scattered light from aerosol particles that are much smaller than precipitation
droplets in the air. Therefore, when the number of aerosol particles that emit scattered light is small, it is difficult to
measure wind information at a distance. Furthermore, as altitude increases, the aerosol density decreases, and the
observation range tends to decrease accordingly. The maximum observation range and aerosol density measured at
each altitude are shown in Inokuchi and Akiyama, 2019.
Advance knowledge of turbulent atmospheric conditions would improve the performance of automatic
aircraft-vibration reduction systems. Automatic control to alleviate aircraft vibration is called gust-alleviation and has
been studied since the 1970s, mostly with only the help of feedback sensors such as inertial measurement units (Regan
and Jutte, 2012). Recently, methods of reducing the vibrations due to turbulence with the help of preview controlling
based on airborne Lidar observation have been reported by Schmitt et al., 2007; Fezans et al., 2019; and Hamada,
2019. The aim of the Aircraft Wing with Advanced Technology Operation (AWIATOR) project is the development
of new direct-lift control devices and a Lidar system for turbulence measurement (Schmitt et al., 2007). Another
project—"Demonstration of Lidar-based CAT detection" (DELICAT) (Barny, 2012)—developed airborne ultraviolet
Lidar for gust and turbulence measurements. The test flights were carried out using an Airbus 340 aircraft equipped
with ultraviolet Lidar. In both the AWIATOR and the DELICAT experiments, the measurement range was short,
because the Lidar was developed for controlling the aircraft automatically.
In order to implement an airborne Doppler Lidar gust-alleviation system successfully, it is very important to
measure the vertical wind velocity accurately. Both horizontal and vertical winds affect aircraft motion, but the effect
of changing the vertical wind velocity is greater. This is because the effect of modifying the angle of attack is relatively
larger than the effect of changing the horizontal wind velocity, which affects only the airspeed (Fezans et al., 2019).
However, a fixed single Doppler Lidar system can only detect the line-of-sight (LOS) wind, providing a one-
dimensional piece of information; the vertical wind velocity in front of the aircraft cannot be measured by such a
system (Hamada, 2019). It is necessary to perform the Lidar measurements in two directions, upward and downward,
to obtain the vertical wind velocity (Neininger, 2017). Figure 1 shows a representation of this concept. The vertical
wind-velocity vector is generated from the differences between the upward and downward LOS winds by using simple
vector conversion. Unfortunately, this method is incapable of estimating the vertical wind velocity with high accuracy
to control the aircraft automatically because the technique assumes homogeneity between the upward and downward
Lidars (Fezans et al., 2019). In this study, a fully turbulent field with atmospheric turbulence and gusts is considered;
under these conditions, it is difficult to estimate the vertical wind velocity with high accuracy using simple vector
conversion. In particular, the estimation accuracy of the vertical wind velocity rapidly worsens when the estimation
position is located farther ahead from the aircraft.
In addition, actual Lidar observations involve errors, noise, and loss of data, with negative effects on aircraft
control, as reported by Misaka et al. (2015); these problems are worse at higher altitudes, where the aerosol density is
smaller than it is at lower ones. Misaka et al. (2015) proposed a filtering algorithm based on a simple Kalman filter to
remove wind-velocity errors from Lidar measurements. For preview control, it is essential to deal with the Lidar errors,
noise and loss of data more carefully. An accurate airflow vector estimation method and an efficient real-time filtering
algorithm are required.
In this study, an estimation method and an airflow-vector filtering algorithm are proposed for preview control
to prevent turbulence-induced aircraft accidents. The method works for both horizontally and vertically directed winds,
and uses both upward and downward Lidars. (In this study, "horizontal wind" means any headwind/tailwind
component that does not include the crosswind component.) The Lidar system in this paper is that also used by JAXA
in its ongoing "Lidar-based gust alleviation control" research project. The Lidars are assumed to be compliant with
the specifications for preview control currently under development by the JAXA. The proposed algorithm stores the
LOS wind data continually and uses recent and past LOS wind data to estimate the airflow vector and the wind field
between Lidars, whereas simple vector conversion utilizes only recent LOS wind data. The airflow vector is calculated
by using wind data extrapolated from the horizontal and vertical wind components; the estimation accuracy of the
airflow vector in front of the aircraft is improved by using such extrapolated wind data because the region between
the Lidars represents a non-homogeneous one. A polynomial expression is used to extrapolate the wind field. In
addition, the proposed method can estimate the two-dimensional distribution of the wind field between the Lidars,
which simple vector conversion cannot.
Two test configurations—an ideal vortex flow field and a weather field—are calculated by an NWP system
and utilized to evaluate the performance of the airflow vector. These experiments generate a large number of pseudo-
Lidar measurements along flight routes from the reference wind field for evaluation of the estimated performance.
Comparing the prediction results with the reference wind field can confirm all the wind-field values.

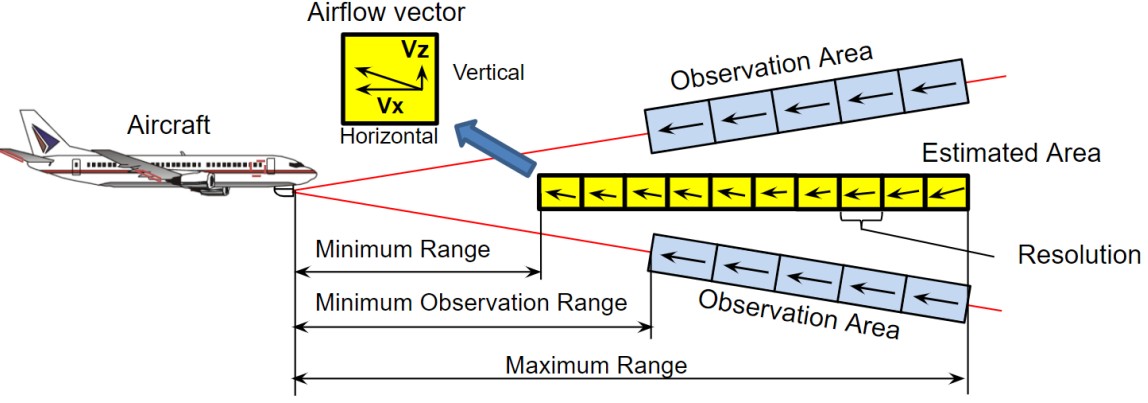


**Fig. 1      Concept of the airborne Lidars observation system**
**2 Methods**

**2.1 Airborne Lidar Specifications**

The airborne Lidar observation system currently under development by JAXA for preview control to prevent
turbulence-induced aircraft accidents is shown in this section. This system has airborne Lidars that are aiming upwards
and downwards; the angle between them is 20 degrees, that is, 10 degrees between the horizontal line and each Lidar.
The Lidar sensor is shown in Fig. 2; its specifications are given in Table 1 (Inokuchi and Akiyama 2019). Laser pulses
generated by an optical transceiver are amplified by optical amplifiers (Sakimura et. al. 2013) incorporated into an
optical antenna and radiated into the atmosphere from optical telescopes. The heat generated by the optical amplifiers
is dissipated by a water-cooled chiller unit. The optical antenna is equipped with a 150 mm large-aperture telescope
for long range observations and a 50 mm small-aperture telescope for vector conversion of short-range observations.
Each Lidar measures the LOS wind velocity with an observational accuracy of $\pm$ .09 m s$^{-1}$; the paired values are used
to estimate the airflow vector in the region between the Lidars. The observational resolution of each Lidar is
approximately 25 m. There are additional performance requirements for preview control: the estimation frequency
and estimation accuracy of vertical wind velocity. The frequency of estimation must be more than 5 Hz, and the
estimation accuracy of the vertical wind velocity must be better than 2.6 m s$^{-1}$ in the LOS distance of 500 m. The
control requirements are the conditions that are necessary for halving the peak variation in acceleration by control.
This value has been specified using control simulations (Hamada, 2019), and Monte Carlo simulations have also been
performed.

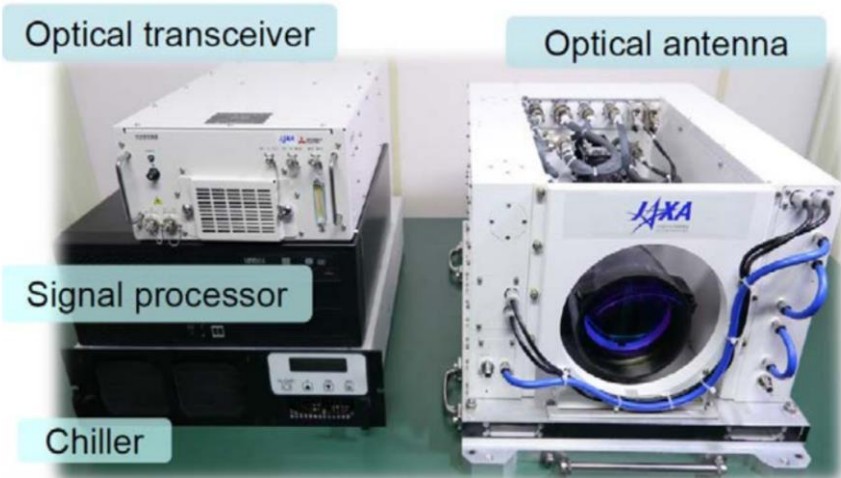


**Fig. 2     Coherent Doppler Lidar used in this work**


**Table 1. Coherent Doppler Lidar Specifications**

| | |
|---|---|
| Laser Wavelength | 1.55 µm |
| Laser Output | 3.3 W |
| Pulse Repetition Frequency | 1,000 Hz |
| Laser Beam Diameter | 150, 50 mm |
| System Weight | 83.7 kg |
| Power Consumption | 936 W |
| Data Rate | 5 Hz |

Next, an existing technique for estimating the airflow vector from a pair of LOS wind values is reviewed.

The airflow vector in the region between the upward and downward Lidars is conventionally estimated via simple
vector conversion. This procedure is similar in concept to the vertical azimuth display approach used in general ground
Lidar systems (Newsom et al., 2017). The simple vector conversion is given by

$$u_x^T = \frac{\left(W_1^T + W_2^T\right)}{2\cos\theta},$$
$$u_z^T = \frac{\left(W_1^T - W_2^T\right)}{2\sin\theta},$$

(1)

where $u_x^T$ and $u_z^T$ are the horizontal and vertical wind velocity measurements at the observation time $T$; $W_1^T$ and
$W_2^T$ are the LOS wind velocities of the upward and downward directed Lidars at the observation time $T$; and $\theta$ is the
angle between the horizontal line and each Lidar, which is 10 degrees in this study. The simple vector conversion
assumes that the wind-field region between the Lidars is homogeneous (Newsom et al., 2017). The assumption of
homogeneity seems natural: the regions between the Lidars are 69.5 m and 173.6 m at the LOS distances of 200 m
and 500 m ahead of the aircraft (Fig. 3). Nevertheless, the assumption would be wrong if a large fluctuation in wind
velocity occurs, creating turbulence. In homogenous conditions, a simple vector conversion can estimate the airflow
vector accurately; however, in non-homogenous conditions, the estimation is expected to have poor accuracy.

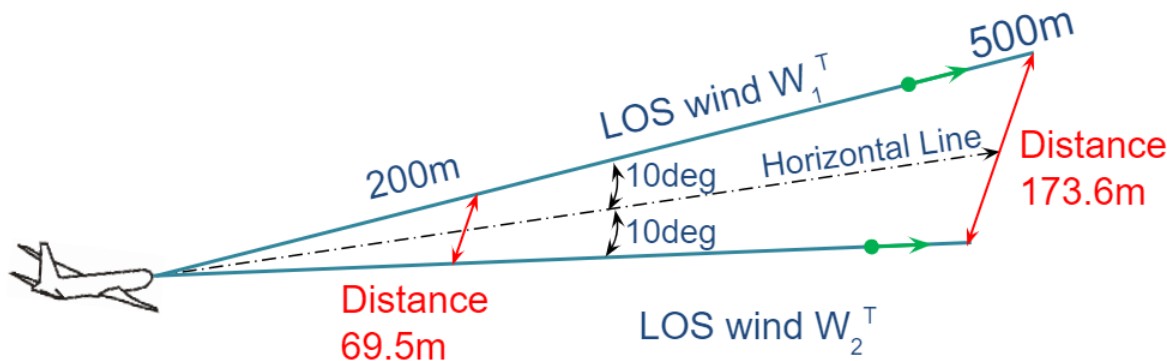


**Fig. 3**    **Distance to wind-field region between the Lidars for two line-of-sight (LOS) distances**

**2.2 Estimation Algorithm Based on Extrapolation**

Whereas simple vector conversion utilizes recent LOS wind data to estimate the airflow vector, our proposed

method stores the LOS wind data continuously and uses both recent and past values to extrapolate the wind field in
the region between the Lidars where it has not been directly measured. The airflow vector is then calculated from Eq.
(1) and the extrapolated horizontal and vertical components of the wind velocity. The airflow-vector estimation
accuracy far ahead of the aircraft is improved relative to simple vector conversion by using the extrapolated wind data
because the region between the upward and downward Lidars is no longer assumed to be homogeneous; our algorithm
uses a polynomial expression to extrapolate data points from both recent and past measurements, allowing it to be
used in non-homogenous wind fields. In addition, the proposed method can estimate the two-dimensional distribution
of the wind field between the Lidars, again unlike simple vector conversion.

Figure 4 shows the overview of the proposed estimation method when a current data point and two past data
points are used. When the aircraft speed is $V$ and the time span of observation is $dt$, the airflow moves backwards at
$V \times dt$ because the aircraft is advancing. Current observation times are denoted as $T$ and past observation times as $T$-1
and $T$-2. The proposed method uses the current LOS wind values ($W_1^T$ and $W_2^T$) and the past LOS wind values ($W_1^{T-1}$
$W_2^{T-1}$ and $W_1^{T-2}$, $W_2^{T-2}$). The perpendicular distances between the horizontal line and each Lidar are denoted as $z^T$,
$z^{T-1}$, and $z^{T-2}$, respectively. Depending on the number of past LOS wind data used, the order of the polynomial
expression used in the extrapolation varies. The aerosol concentration in the upper sky is low, suggesting that there is
considerable missing data and noise. A sufficient number of past LOS wind data may not be available to estimate a
high-order polynomial expression, and this could affect the robustness of the control. For this reason, a first-degree
polynomial expression is adopted in this study and used in the least-squares method (LSM) to extrapolate the wind-
field values according at the horizontal line. The airflow vector is calculated by Eq. (1) using the extrapolated LOS
wind. The equation used in the extrapolation method is

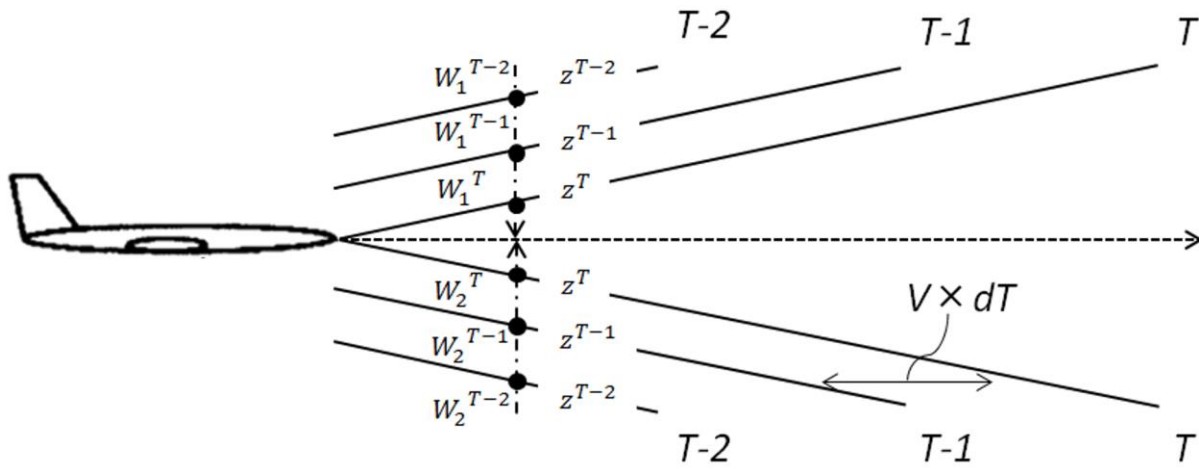


**Fig. 4     Overview of estimation by proposed method when line-of-sight wind data from 0, 1, and 2 past time-steps dT**
**are used. V = speed of aircraft; $W_1^T$ and $W_2^T$ = wind speeds measured at time T by the two Lidars; z = vertical distance**
**perpendicular to velocity of aircraft**

$$W_j'(z) = a_j z + b_j \ , \tag{2}$$

    where

$$a_j = \frac{N \sum_{i=T-(N-1)}^{T} z^i W_j^i - \sum_{i=T-(N-1)}^{T} z^i \sum_{i=T-(N-1)}^{T} W_j^i}{N \sum_{i=T-(N-1)}^{T} (z^i)^2 - \left(\sum_{i=T-(N-1)}^{T} z^i\right)^2},$$

(3)

$$b_j = \frac{\sum_{i=T-(N-1)}^{T} z^i \sum_{i=T-(N-1)}^{T} W_j^i - \sum_{i=T-(N-1)}^{T} z^i W_j^i \sum_{i=T-(N-1)}^{T} z^i}{N \sum_{i=T-(N-1)}^{T} (z^i)^2 - \left(\sum_{i=T-(N-1)}^{T} z^i\right)^2}.$$


**2.3 Filtering Error and the Lack of Wind-Velocity Data**

In this study, two filtering algorithms are used to remove the error and the loss of data in airborne Lidars.

First, a filtering algorithm that is a simple representation of a Kalman filter with simplified Kalman gain is used; this
filtering algorithm is described in detail in the study of Misaka et al., 2015. The algorithm assumes that infinite
variance is used to exclude outliers and loss of data. This method uses the Lidar spectrum data at each range-bin; the
algorithm defines the validity of the measurements during the Lidar data peak-detection process. To identify the
correct and incorrect LOS wind-velocity values, two spectrum thresholds are defined. First, the largest and second-
largest spectrum values, $k_{1st}$ and $k_{2nd}$, which are the Fast Fourier Transform points for the first and second spectrum
peaks, respectively, are adjacent to each other; i.e., the magnitude of the distance between the largest and second-
largest spectrum values in the Fast Fourier Transform is equal to one. Second, the distance between $k_{1st}$ and the
averaged spectrum peak $k_{ave}$ is required to be less than a certain value $k_{dif}$, which represents the only hyper-parameter
in this algorithm as well as a parameter related to smoothness. $k_{ave}$ is the index that conveys the location of the spectrum
peak averaged in short ranges, e.g., 2–30 range-bins from the lidar origin. Figure 5 shows a conceptual explanation of
the variables of simplified Kalman gain in the cases of correct measurement and of an error peak. In this study, the
filtering algorithm is carried out first when the observation data is obtained:

$$K = \begin{cases} 1 & |k_{1st} - k_{2nd}| = 1 \ and \ |k_{1st} - k_{ave}| < k_{dif} \\ 0 & Otherwise \end{cases}$$

(4)

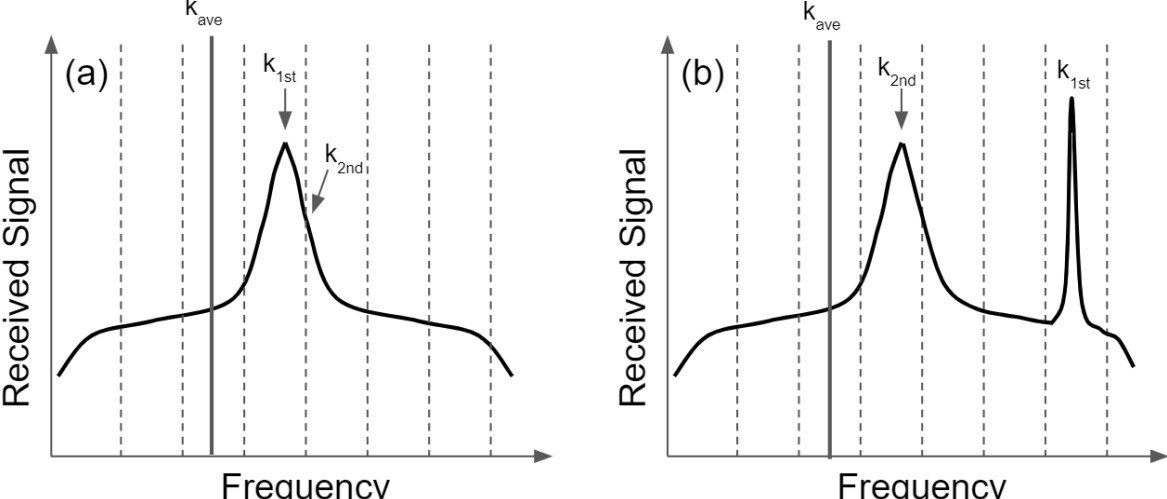


**Fig. 5     Conceptual explanation of the variables of simplified Kalman gain.**
**(a) Correct measurement case of *K*=1.  (b) Case with the error peak of *K*=0**

Secondly, a robust least-squares estimation, based on Tuckey's biweight methodology (Huber, 2008), is
carried out to reduce the impact of the error in the LOS wind velocity. This method is based on the LOS wind data, in
contrast to the spectrum data from Lidar observations in the first method. Although the filtering algorithm based on a
simple Kalman filter can remove the error from the Lidar spectrum data, error filtering via this algorithm is not perfect
despite being useful. As error data can be a reason for miscontrol, it is essential to deal with the error and the loss of
data of the Lidars more carefully when the filtering algorithm is used for the preview control. Therefore, the robustness
of the estimated airflow vector is secured by combining the simple Kalman filtering algorithm with the results of
robust LSM, using Eqs. (2) and (3). In addition, the robust LSM estimation can employ the extrapolation algorithm
effectively as per Eqs. (2) and (3). Therefore, a simpler and more robust algorithm is provided. Figure 6 explains the
concept behind Tuckey's biweight methodology as applied to Lidar. The fundamental principle involves comparing
the observed LOS wind values with the estimated ones from the polynomial expression used in the LSM. In the 1st
step, the LOS wind is estimated using the general LSM (Eq. (2)). In the 2nd step, the difference $d_j^T$ between the
observed LOS wind value and that estimated from the polynomial expression is found:

$$d_j^T = W_j^T - \left( a_j z + b_j \right). \tag{5}$$

A permissible difference range $L$ is defined and weights $w_j^T (d_j^T)$ are calculated depending on where $d_j^T$ falls in the
distance range:

$$w_j^T(d_j^T) = 0 \ \left( d_j^T < -L \right)$$

$$w_j^T(d_j^T) = \left( 1 - \left( \frac{d_j^T}{w_j^T} \right)^2 \right)^2 \left( -L \le d_j^T \le L \right) . \tag{6}$$

$$w_j^T(d_j^T) = 0 \ \left( d_j^T > L \right)$$

Weights are assigned to each LOS wind velocity value. In the 3rd step, a new first-degree polynomial expression for
the LSM with the weighted data is estimated as follows.

$$a_j' = \frac{\sum_{i=T-(N-1)}^{T} w_j^i \sum_{i=T-(N-1)}^{T} w_j^i z^i W_j^i - \sum_{i=T-(N-1)}^{T} w_j^i z^i \sum_{i=T-(N-1)}^{T} w_j^i W_j^i}{\sum_{i=T-(N-1)}^{T} w_j^i \sum_{i=T-(N-1)}^{T} w_j^i (z^i)^2 - \left(\sum_{i=T-(N-1)}^{T} w_j^i z^i\right)^2}$$

$$b_j' = \frac{\sum_{i=T-(N-1)}^{T} w_j^i z^i \sum_{i=T-(N-1)}^{T} w_j^i W_j^i - \sum_{i=T-(N-1)}^{T} w_j^i z^i W_j^i \sum_{i=T-(N-1)}^{T} w_j^i z^i}{\sum_{i=T-(N-1)}^{T} w_j^i \sum_{i=T-(N-1)}^{T} w_j^i (z^i)^2 - \left(\sum_{i=T-(N-1)}^{T} w_j^i z^i\right)^2}$$

(7)

This process is repeated until the weight of the error value decreases and converges.

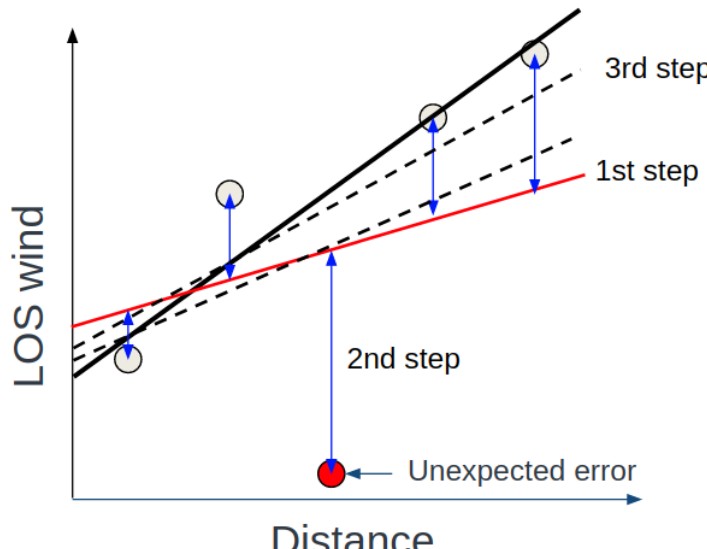

**Fig. 6 Conceptual explanation of Tuckey's biweight methodology applied to line-of-sight (LOS) wind at various distances.**
**First step: simple least-squares fit. Second step: observations are compared with the estimate. The data are weighted, and**
**extreme outliers are excluded, using Eq. (6). Third step: Least-squares fit of the weighted data.**



**2.4 Filtering Wind-Velocity Noise**

Lidar is subject not only to measuring errors and loss of LOS data values but also to random noise; this type
of noise also leads to a poor estimation of the airflow vector. The random noise is caused by the reduced intensity of
the received light due to the thin aerosol concentration in the sky. A general Lidar signal consists of random noise
superimposed on the spectral signal. If the signal intensity is low, peak search may only detect the random noise.
(Additional randomness caused by environmental factors and data processing in Lidar is considered here as
randomness of the wind-speed values.)
A simple spline algorithm generates a curve that passes through all sample points; therefore, it is not able to
generate a smooth curve when the sample points have random noise, and a smoothing spline algorithm is often applied
to remove the random noise in the Lidar LOS wind values, as in the study by Woltring, 1986. The curve generated by
this algorithm does not pass through all sample points, and because of that, it can be smoother, even when there is
random noise from Lidar LOS wind measurements. The smoothing spline model minimizes the criterion function $C_p$,

$$C_p = \sum_{i=1}^{n} \quad v_i \{y_i - s_p(x)\}^2 + p \int \quad \left(\frac{d^2 s_p}{dx^2}\right)^2 dx , \tag{8}$$


where $y_i$ is a sample point value, $s_p(x)$ is the value generated by a simple spline algorithm, $v_i$ is a weighted factor, and
$p$ is the regularization parameter. The smoothest curve is generated when the criterion function $Cp$ is minimized.

**2.5 System Flowchart**

The airflow-vector estimation algorithm is a sequence of five different processes, which are summarized
below. The system flowchart is shown in Fig. 7.
1)    The filtering algorithm based on a simple Kalman filter is used to remove the error in Lidar LOS
wind-data values.
2)    The smoothing spline method is applied to reduce the negative effect of the random noise in LOS
wind-data values and extrapolates the values at positions for which no measurements can be read. This is identified
as the first-step error.
3)    Extrapolation, based on the polynomial expression, is carried out to estimate the wind-field values
by using current and past LOS wind data.
4)    A robust LSM model is applied to obtain a more accurate polynomial expression. The calculation
repeats until the parameter converges.
5)    The airflow vector is calculated by Eq. (1) with the extrapolated LOS wind.

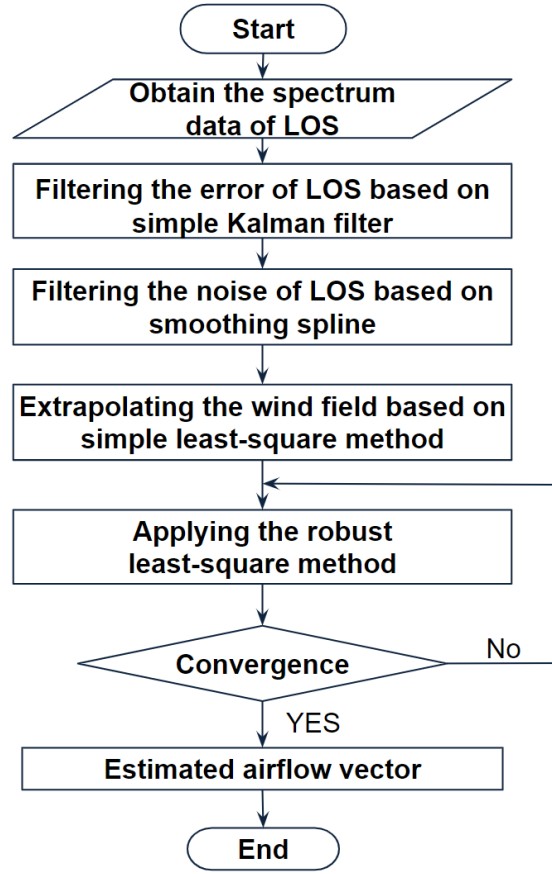


**Fig. 7 System flowchart for the airflow vector estimation algorithm**

**3 Test Configurations**

**3.1 Ideal Vortex Model**

We have conducted numerical experiments to evaluate the performance of actual airborne Lidars. The ideal

vortex model is defined and used to evaluate the estimated performance of the airflow vector. In this study, the
Hallock-Burnham vortex model (Hinton et al., 1997) is used. The experiment generates a large number of pseudo-
Lidar values, from which the airflow vector is estimated. The estimation results are then compared with the reference
wind-field values of the ideal vortex model. Figure 8 shows the distribution of wind velocity generated using the
Hallock-Burnham vortex model.

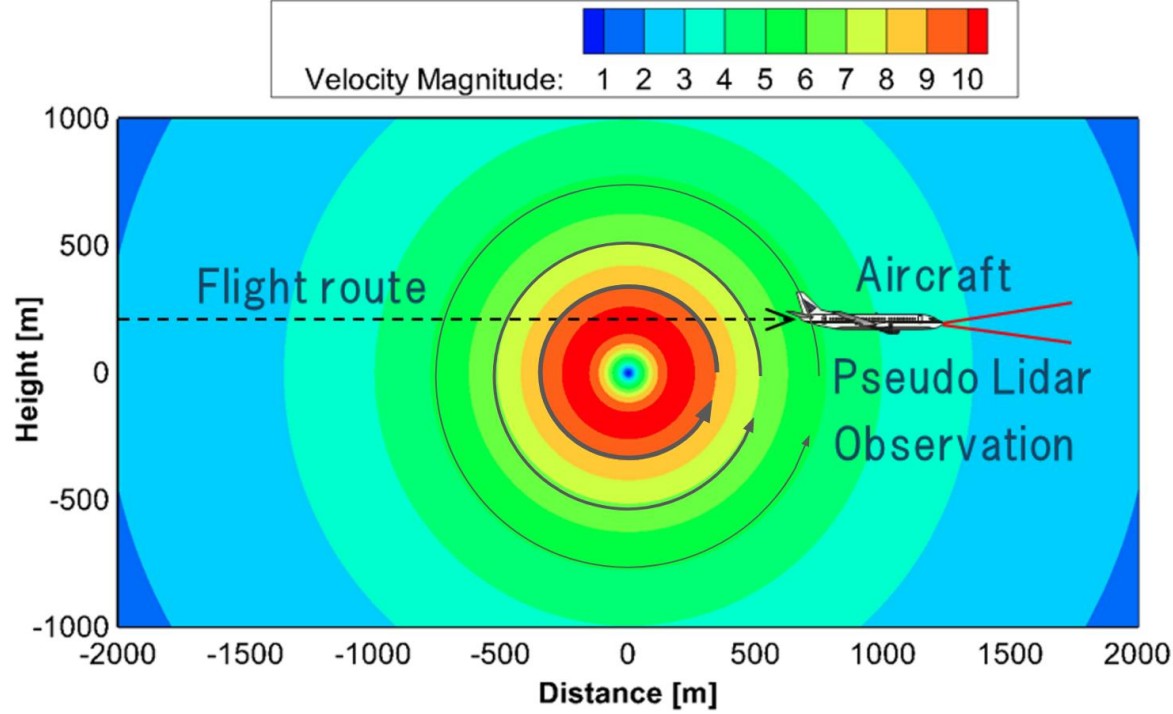

Fig. 8    The distribution of vertical wind velocity generated by the Hallock-Burnham vortex model

**3.2 NWP model**

The results predicted by a numerical weather model—the Japan Meteorological Agency Non-Hydrostatic Model (JMA-NHM)—are used to evaluate airflow-vector estimation performance (Saito et al., 2007; Kikuchi et al., 2015). To obtain high-resolution weather prediction, a one-way multi-nesting technique (Kikuchi et al. 2015) is employed for downscaling purposes. The computational domain is nested four times to increase grid resolutions from 5.0 to 0.05 km gradually (in the sequence 5.0, 1.5, 0.5, 0.15, and 0.05 km).

Three-hour mesoscale objective analysis data, collected using a mesoscale four-dimensional variational data-assimilation system at the Japan Meteorological Agency (Saito et al., 2007), are used for the initial condition of 5.0 km grid resolution. The experiment generates a large number of simulated twin-Lidar observation values along flight routes from the wind-field data generated by JMA-NHM, which are more realistic than ideal-vortex model results. The airflow vector is estimated from the pseudo-Lidar observations and compared with the JMA-NHM reference field. Figure 9 shows the distribution of the vertical wind velocity values generated by JMA-NHM.

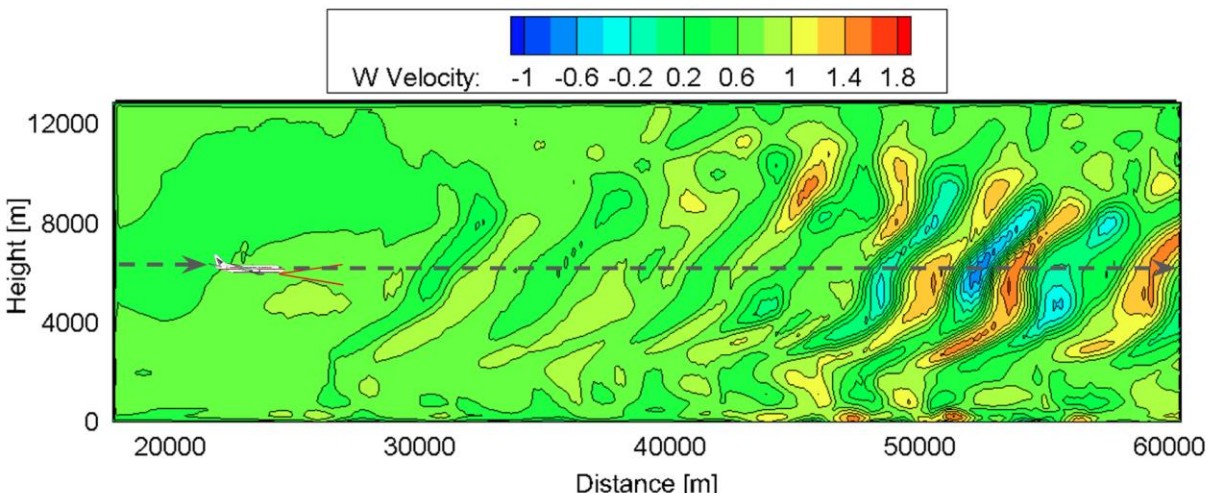

**Fig. 9**      **Vertical wind velocity distribution map generated by JMA-NHM**
**3.3 Generation of pseudo-errors and noise**
To confirm the effectiveness of the proposed filtering algorithms, errors and noise are generated artificially
by using the parameter of the backscattering coefficient in the atmosphere and the statistics-based coherent Lidar
equation (Kameyama et al., 2007). The backscattering coefficient is strongly related to the aerosol density in the
atmosphere, and it has an impact on the Lidar measurements and estimation performance. When the backscattering
coefficient is very low, the measurement performance is worse, and the LOS wind data show errors and noise. Apart
from this, the measurement performance is related to the focal distance, pulse width, and Lidar power (Kameyama et
al., 2007). The signal-noise ratio (SNR) at the receiver, at each LOS distance, is calculated by using the coherent Lidar
equation and the detailed operating condition of JAXA's Lidar (Inokuchi and Akiyama 2019):

$$SNR(R) = \frac{\eta\, P_t\, \Delta R\, \beta\, K^{2R}\, \frac{\pi D^2}{4R^2}}{h\, f\, B\, SRF(R)} \tag{9}$$

$$SRF(R) = 1 + \left\{1 - \frac{R}{F}\right\}^2 \left\{\frac{k(A_c D)^2}{8R}\right\}^2 + \left\{\frac{A_c D}{2S_0(R)}\right\}^2 \tag{10}$$

$$S_o(R) = (1.1\, k^2\, R\, C_n^2)^{-\frac{3}{5}} \tag{11}$$


Here, $R$ is the observation distance, $\eta$ is the system efficiency, $P_t$ is the light-transmission power, $\Delta R$ is the resolution
range, $\beta$ is the backscattering coefficient, $K$ is the atmospheric transmittance, $D$ is the opening size of the optical
antenna, $h$ is Planck's constant, $f$ is optical frequency, $B$ is received bandwidth, $F$ is focal distance, $k$ is wave number,
$A_c$ is the vignetting factor of the optical antenna, and $C_n^2$ is the atmospheric structure constant. In this study, the
conditions are set according to the design specification for airborne Lidars. Six atmospheric conditions are prepared
in order to evaluate the filtering performance. The backscattering coefficients are (standard case) $1.8\times10^{-8}$ sr$^{-1}$m$^{-1}$, (a)
$1.8\times10^{-11}$ sr$^{-1}$m$^{-1}$, (b) $1.35\times10^{-11}$ sr$^{-1}$m$^{-1}$, (c) $0.9\times10^{-11}$ sr$^{-1}$m$^{-1}$, (d) $0.45\times10^{-11}$ sr$^{-1}$m$^{-1}$, and (e) $0.18\times10^{-11}$ sr$^{-1}$m$^{-1}$. Figure
10 shows the statistics for the error and noise as functions of SNR bandwidth.
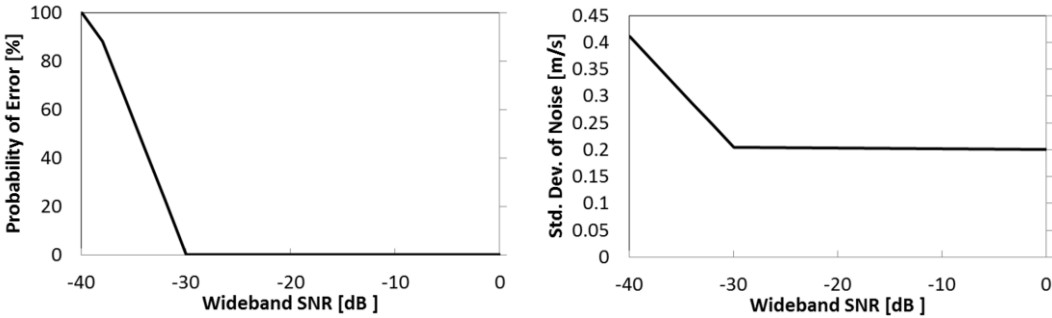
**Fig. 10** **Probability of error and standard deviation of noise as functions of signal-noise ratio (SNR)**
**bandwidth**
**4 Results**
**4.1 Ideal Vortex Model without Error and Noise**

The numerical experiments with the ideal vortex model have been carried out, and Figs. 11 and 12 show the

distributions of the horizontal and vertical wind components that are estimated by the simple vector conversion and
the proposed method. The flights start at the edge of the computational space. Figs. 11 and 12 show the results after
10 and 15 s, respectively. Thus, they represent the instants of time before and during the aircraft's close approach to
the vortex core. As shown in Figs. 11 and 12, the simple vector conversion method, which assumes that the wind field
of the region between the Lidars is homogeneous, cannot accurately reproduce the two-dimensional distribution
between the Lidars. On the other hand, the figures confirm that the proposed method can estimate the two-dimensional
distribution of wind-field values between the Lidars. Figure 11 shows that the two-dimensional distribution obtained
with the proposed method is very similar to that of the reference field. In addition, the results show that the horizontal
wind velocity with simple vector conversion is approximately –7 m/s, whereas that with the proposed method is –9.5
ms$^{-1}$; the horizontal wind velocity of the reference field is –9.0 ms$^{-1}$ at LOS distance of 450–500 m. Figure 12 shows
that the results of the horizontal and vertical wind velocities with simple vector conversion are considerably lower
than those of the reference field. The horizontal wind results show that the value obtained with the simple vector
conversion is approximately –9.5 ms$^{-1}$, whereas that with the proposed method is approximately –3.5 ms$^{-1}$; the
horizontal wind velocity of the reference field is approximately –4.5 ms$^{-1}$ at LOS distance of 450–500 m. The vertical
wind results show that the value obtained with simple vector conversion is approximately –1.0 m/s, whereas that
obtained with the proposed method is approximately 8.5 ms$^{-1}$; the vertical wind velocity of the reference field is
approximately 7.0 ms$^{-1}$ at LOS distance of 450–500 m. Therefore, simple vector conversion has significantly large
errors between the reference and estimated values. The errors in both the horizontal and vertical wind values estimated
by the proposed method are much smaller than those estimated with simple vector conversion. Although the two-
dimensional distribution of the horizontal wind-field values of the proposed method is larger than that of the reference
field at a LOS distance of 450–500 m, the vertical wind-field values can provide a good assessment of the reference
field shown in Fig. 12. The 15 s timing in Fig. 12 is a more challenging case than others because the aircraft is
positioned very close to the center of the vortex, and the wind direction changes abruptly. Although it is difficult to
estimate the perfect wind-field value at this time by using the proposed method, the proposed estimation method
demonstrably has a much higher accuracy than simple vector conversion. Overall, the proposed method has much
better performance than the simple vector conversion method, and it can estimate the two-dimensional distribution of
wind field values accurately, unlike the simple vector conversion method.
Next, the statistical estimation performance is evaluated using 100 pseudo-routes that are randomly generated
750 m above and below the center of the vortex core; Fig. 13 shows the results for the vertical wind values, along with
the performance required for automatic control. The root mean square error (RMSE) between the reference-field value
and the estimated wind-field value is used for evaluating the estimation performance. Moreover, the effect of the
number of past Lidar observations used to determine the wind field, i.e., the past LOS wind, is checked. Simple vector
conversion cannot satisfy the performance requirement at a LOS distance greater than 350 m. This means that
achieving preview control using the simple vector conversion method may be difficult. At a LOS distance of 500 m,
the RMSEs of the vertical wind values of the simple vector conversion and proposed methods are approximately 4.0
ms$^{-1}$ and 1.2 ms$^{-1}$, respectively. The proposed method can cater to the performance demand even if the number of past
LOS wind values used is different; a lower number leads to better estimation performance.

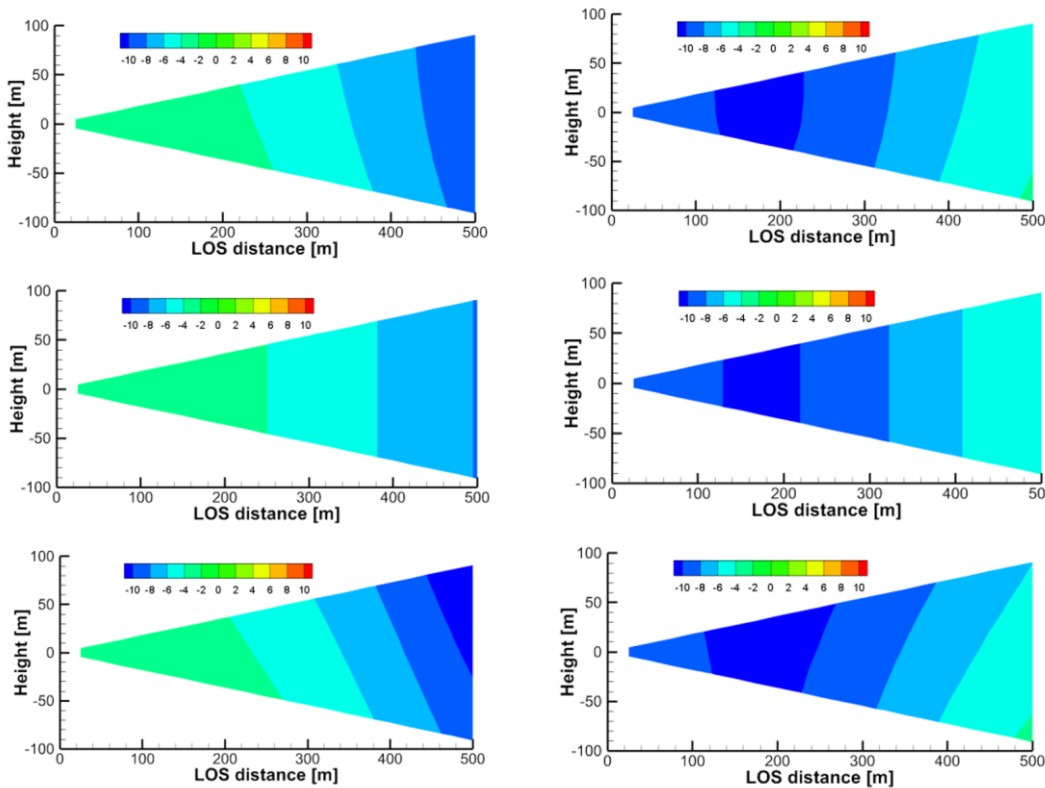


**Fig. 11   Distributions of the horizontal and vertical wind components estimated by the simple vector conversion method**
**vs. the proposed method (at time 10 s). Upper figures: ideal vortex model; middle figures: simple vector conversion**
**method; lower figures: proposed method with five-past LOS wind datasets. Left figures: horizontal wind values; right**
**figures: vertical wind values**

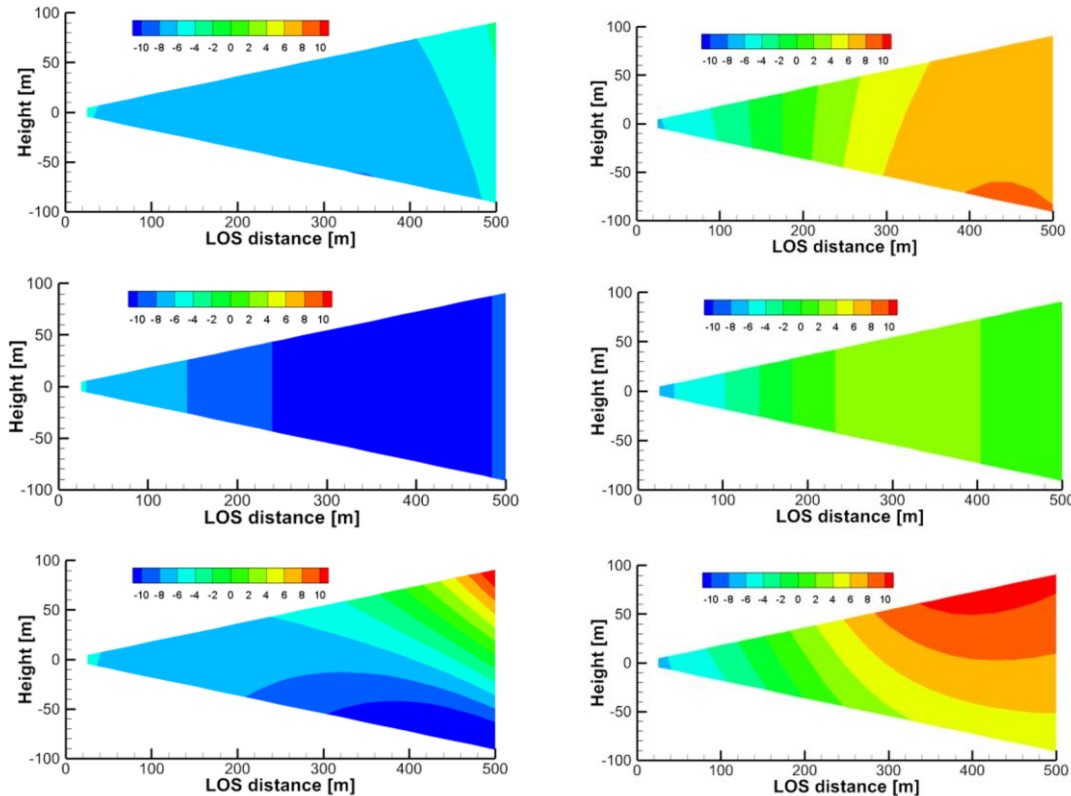


**Fig. 12** **Distributions of the horizontal and vertical wind components estimated by the simple vector conversion method vs. the proposed method (at time 10 s). Upper figures: ideal vortex model; middle figures: simple vector conversion method; lower figures: proposed method with five-past LOS wind datasets. Left figures: horizontal wind values; right figures: vertical wind values**

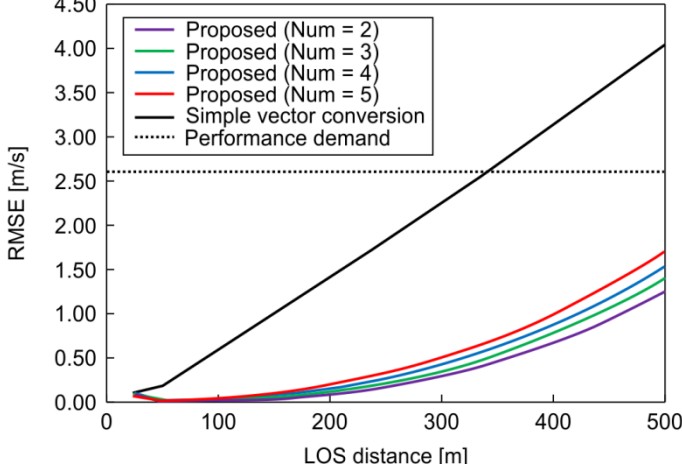

**Fig. 13** **Statistical estimation performance (root mean square error) of vertical wind values (ideal vortex model). Num = number of past line-of-sight (LOS) wind values used**

**4.1 Numerical Weather Prediction without Error and Noise**

We also conducted numerical experiments with NWP values. Figs. 14 and 15 show the distributions of the horizontal and vertical wind components that are estimated by simple vector conversion and the proposed method.

Figure 14 shows the results for the instants of time before and during the approach to a vertical wind fluctuation. The
simple vector conversion method cannot accurately reproduce the two-dimensional distribution of the wind field
between the Lidars. On the other hand, the proposed method can estimate the two-dimensional distribution of the wind
field between the Lidars more accurately. Figure 15 shows that the wind velocities predicted by the simple vector
conversion method are higher than the reference fields at 300-500 m of LOS distance, in contrast to those of the
proposed method.

Next, the statistical estimation performance is evaluated using 100 pseudo-routes that are randomly generated
between 2 km and 10 km altitude. Fig. 16 shows the results, along with the performance requirement for automatic
control. The effect of the number of past LOS wind-values used is also checked. In this case, both simple vector
conversion and the proposed method can satisfy the performance demand for preview control; however, the
performance results of simple vector conversion are much worse than those of the proposed method. Moreover, the
proposed method can estimate quite accurate wind-field values. In this case, the use of a higher number of past LOS
wind values leads to better estimation performance.

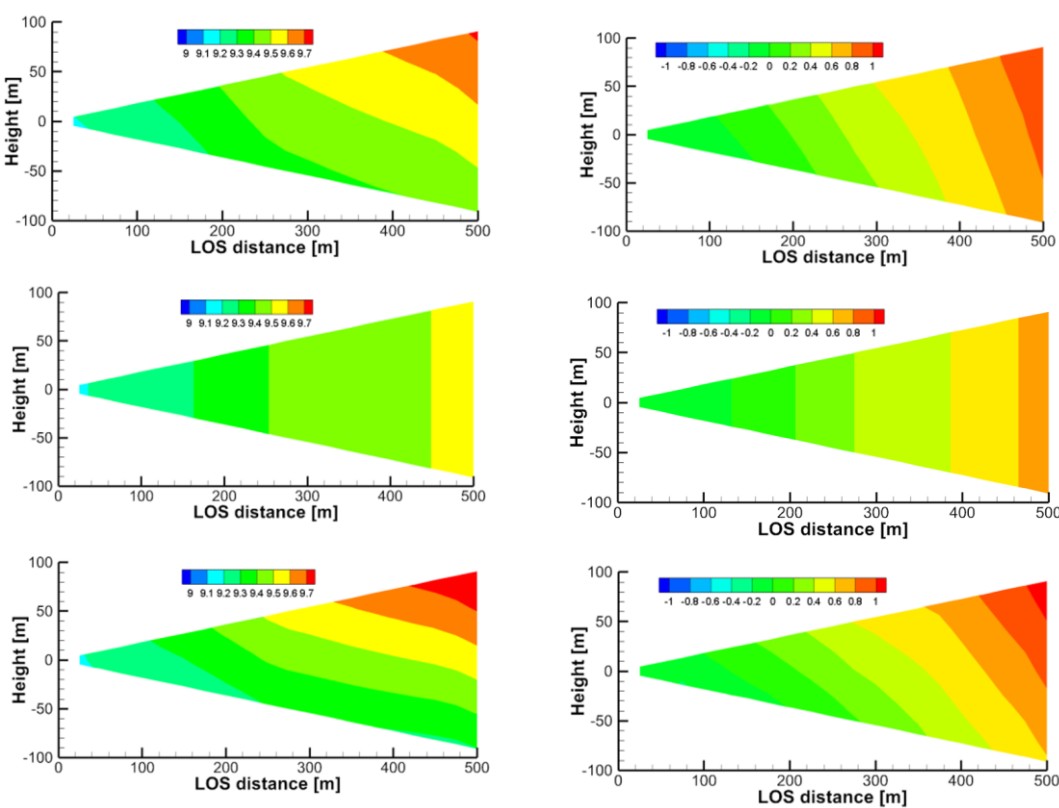

**Fig. 14 Distributions of horizontal and vertical wind components estimated via simple vector conversion and proposed**
**method before approach to vertical wind fluctuation. Upper figures: ideal vortex model; middle figures: simple vector**
**conversion method; lower figures: proposed method with five-past LOS wind datasets. Left figures: horizontal wind**
**values; right figures: vertical wind values**


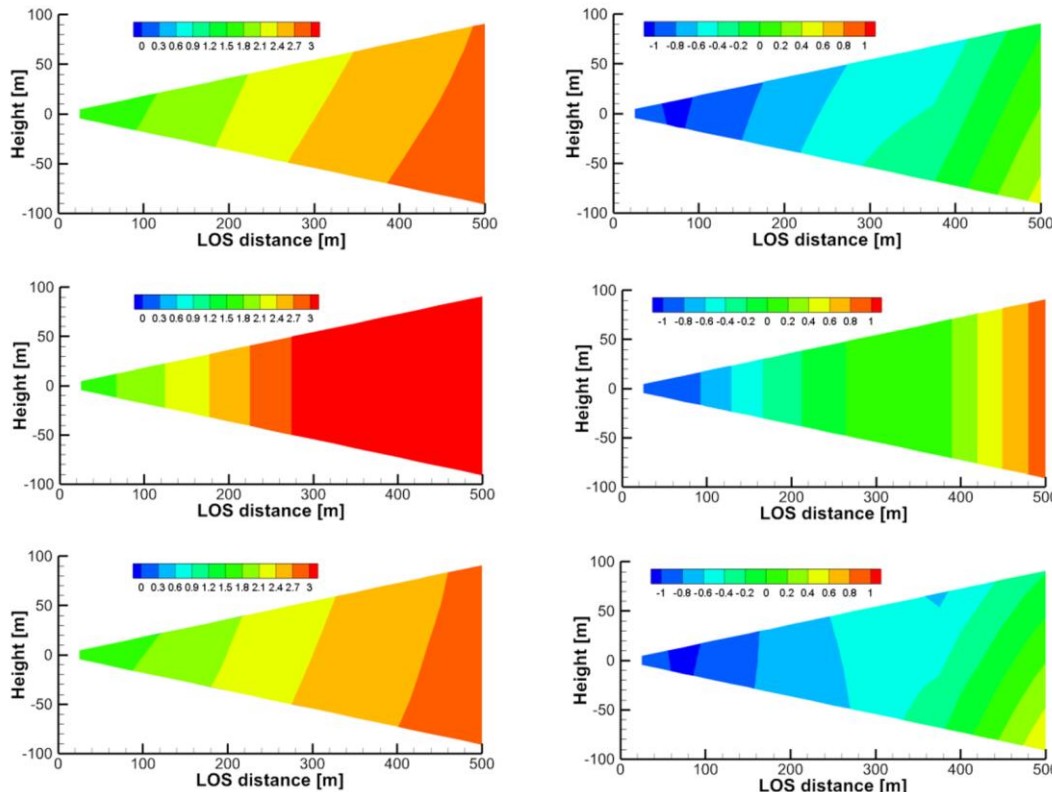

**Fig. 15 Distributions of horizontal and vertical wind components estimated via simple vector conversion and proposed**
**method immediately during approach to vertical wind fluctuation. Upper figures: ideal vortex model; middle figures:**
**simple vector conversion method; lower figures: proposed method with five-past LOS wind datasets. Left figures:**
**horizontal wind values; right figures: vertical wind values**

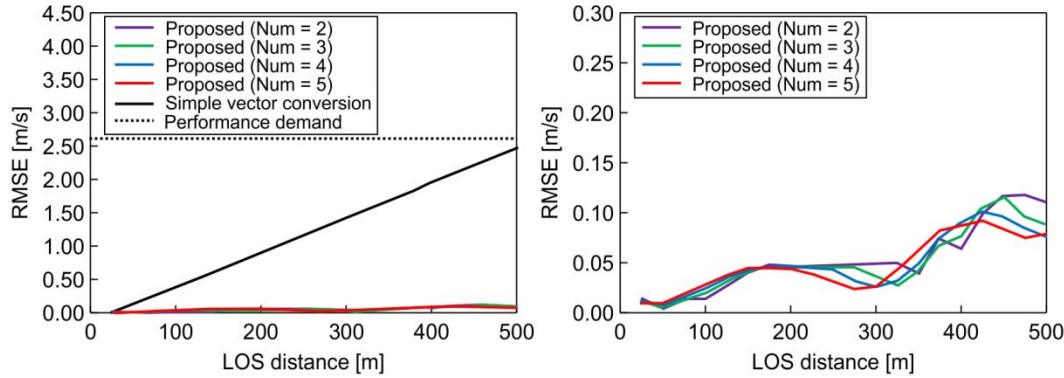

**Fig. 16 Statistical estimation performance (root mean square error) for numerical weather prediction results. Num =**
**number of past line-of-sight (LOS) wind values used**
**4.2 Ideal vortex model with error and noise**
In this section, numerical experiments with error and noise in LOS wind values are conducted to evaluate the
estimation performance of the proposed method. These numerical experiments show the error/noise-filtering
performance difference between simple vector conversion and the proposed method with extrapolation from the past
LOS wind. Six atmospheric conditions are prepared in order to evaluate the filtering performance. The backscattering
coefficients are (standard case) $1.8\times10^{-8}$ sr$^{-1}$m$^{-1}$, (a) $1.8\times10^{-11}$sr$^{-1}$m$^{-1}$, (b) $1.35\times10^{-11}$ sr$^{-1}$m$^{-1}$, (c) $0.9\times10^{-11}$ sr$^{-1}$m$^{-1}$, (d)
$0.45\times10^{-11}$ sr$^{-1}$m$^{-1}$, and (e) $0.18\times10^{-11}$ sr$^{-1}$m$^{-1}$.

First, numerical experiments with the ideal vortex model are carried out. Figure 17 shows the LOS wind

values, which include the measured data with error and noise, the reference wind, the smoothing spline, and the general
spline model results. Figure 17 shows that the smoothing spline can filter the error and noise data of LOS wind values.
When the general spline is used, the error can be filtered correctly by using a simple Kalman filter and a robust LSM;
however, the noise cannot be filtered. Next, the statistical estimation performance is evaluated using 100 pseudo-
routes that are randomly generated 750 m above and below the center of the vortex core. Fig. 18 shows the results of
the statistical estimation performance with error and noise. In addition, the difference due to the atmospheric
conditions in the six cases with different backscattering coefficients is also checked. Simple vector conversion cannot
satisfy the performance demand at a distance farther than 350 m LOS and cannot work correctly under atmospheric
condition (e). The proposed method can always satisfy the performance demand except under atmospheric condition
(e). It thus shows much better performance than simple vector conversion, even though it is difficult to estimate the
wind field values by either method for atmospheric condition (e), which contains much larger noise levels than the
other conditions.

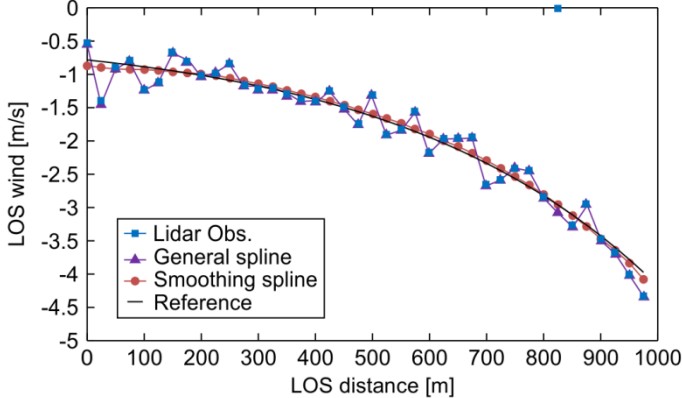


**Fig. 17 Line-of-sight (LOS) wind values: measured data with error and noise, reference wind, smoothing spline, and**
**general spline**

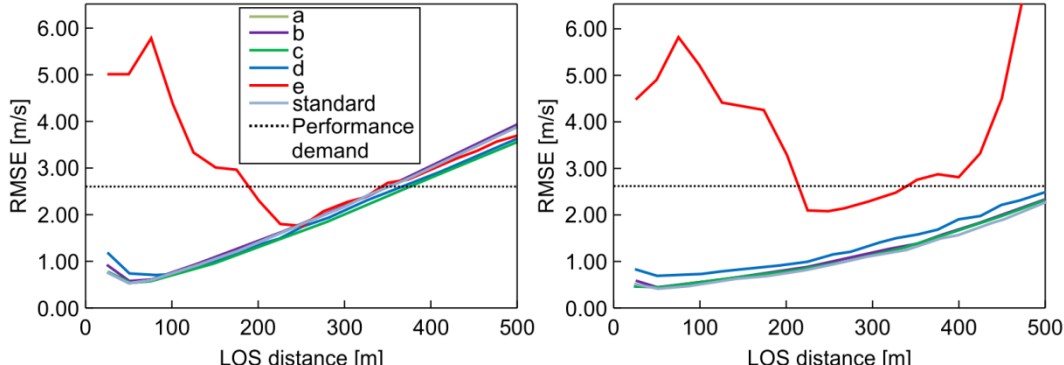


**Fig. 18 Statistical estimation performance (root mean square error) for line-of-sight (LOS) wind velocities including error**
**and noise under six atmospheric-condition scenarios (a–e and standard) (assuming ideal vortex model). Left figure:**
**simple vector conversion; right figure: proposed method**
In addition, the cross-plots of the reference and the estimated vertical wind are shown as Fig. 19. In Figs. 19
(a) and (b) the results of the simple vector conversion are presented; (c) and (d) show the results of the proposed
method. Figs. 19 (a) and (c) are the cases without error and noise, whereas (b) and (d) are the cases with error and
noise. By comparing (a) and (c), we can deduce that the proposed method provides a much better estimation than does
simple vector conversion. The results in (b) and (d) are spread wider than those in (a) and (c), because of the noise
data of LOS wind values. It is worth mentioning that the noise data have more negative effects on the result at 500 m
LOS distance than at 100 m and 300 m LOS. Nevertheless, comparison of (b) and (d) shows that the proposed method
can provide more accurate estimations than the simple vector conversion method.

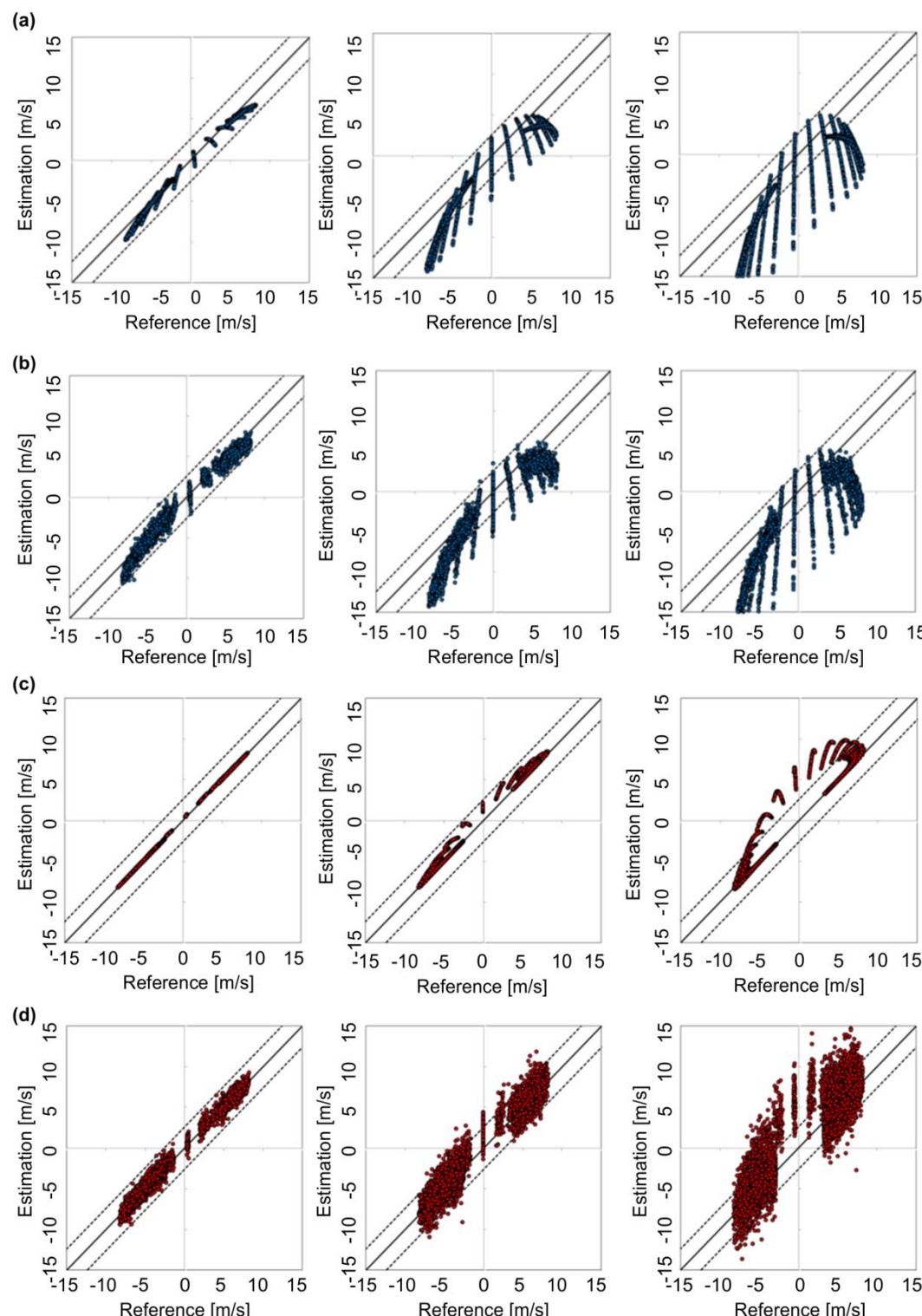


**Fig. 19 Cross-plots of the reference and the estimated vertical wind data. Left figures: 100 m line-of-sight (LOS) distance; middle figures: 300 m LOS distance; right figures: 500 m LOS distance. (a), (b): Simple vector conversion; (c), (d): proposed method. (a), (c): cases without error and noise; (b), (d): cases with error and noise. The dots indicate the wind speed estimated at 5 Hz, and the dotted lines indicate the performance demand for control.**

442

**4.3 Numerical weather prediction with error and noise**

We also carry out numerical experiments with NWP. The statistical estimation performance is conducted by using 100 pseudo-routes between 2 km and 10 km altitude. Fig. 20 shows the results of the statistical estimation performance with error and noise. Six different atmospheric conditions (standard, (a), (b), (c), (d), and (e), defined by their backscattering coefficients) are used. In this case, both simple vector conversion and the proposed method can satisfy the performance requirement for preview control; however, the simple vector conversion shows worse performance than the proposed method. The proposed method can estimate wind-field values quite accurately and displays better performance than the simple vector conversion method. As in the previous experiment, it is difficult to estimate the wind field-values for atmospheric condition (e) by using either simple vector conversion or the proposed method.

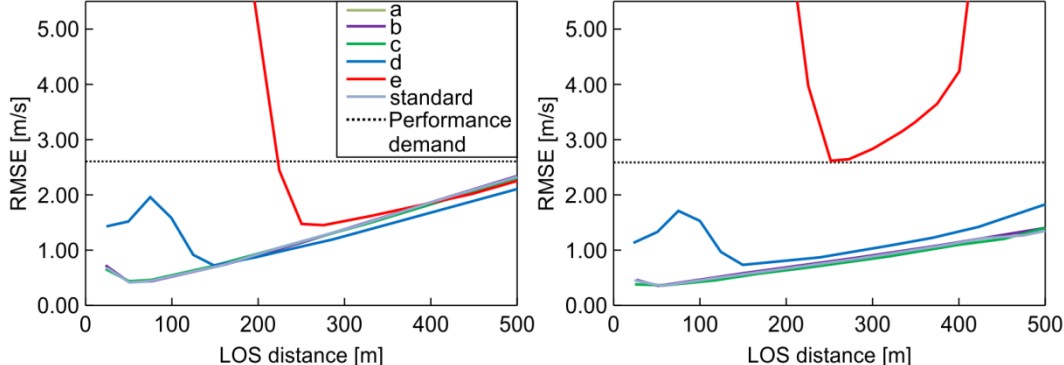

**Fig. 20 Statistical estimation performance (root mean square error) for line-of-sight (LOS) wind velocities (including error and noise) under six atmospheric-condition scenarios (a–e and standard) with numerical weather-prediction data. Left figure: simple vector conversion; right figure: proposed method**

**5. Conclusion**

In this study, an airflow vector estimation algorithm based on upward and downward airborne Lidars has been proposed for preview control to prevent turbulence-induced aircraft accidents. This estimation algorithm uses the technique of extrapolating the wind-field values by using the LSM and the current and past LOS wind datasets to improve the accuracy of estimated wind values. Two test configurations for numerical experiments (ideal vortex flow and realistic NWP weather field values) have been used to evaluate the estimation of the airflow vector.

Numerical experiments on LOS wind estimation show that the proposed extrapolation method has much better performance than simple vector conversion methods, and it can estimate the two-dimensional distribution of wind-field values accurately, which simple vector conversion cannot. The estimation performance and the computational cost of the proposed method can satisfy the performance demand for preview control.

Numerical experiments with error and noise in the LOS wind data have been conducted to evaluate the performance of the proposed estimation method. These numerical experiments show that the smoothing spline model can filter noise correctly and reduce its negative effects. The proposed method performs much better than the simple vector conversion method, although it is difficult to estimate the wind-field values for atmospheric condition (e) with either method. Atmospheric condition (e) has more noise than other conditions, and when the noise exceeds a certain level, it becomes difficult to estimate the air flow regardless of the method applied.

The proposed algorithm can satisfy the performance demands for preview control in both estimation
performance and computational cost. It can estimate a two-dimensional distribution that cannot be estimated by
existing methods. This is valuable for improving the accuracy of the preview control: for example, the proposed
method can cope with the critical case where the flight direction of the aircraft is at a steep angle with the aircraft
either ascending or descending.
The findings of this study are subject to certain limitations. The target size of the atmospheric turbulence is
assumed by the proposed algorithm to be comparable to or larger than the observation region between the Lidars.
Therefore, it is difficult to estimate a wind field with turbulence smaller than this. The effect on the aircraft vibration
due to such minor turbulence, however, is minimal. An exception to this is aircraft-generated wake turbulence, which
still poses a safety risk. The radius of the actively fluctuating wake-turbulence core is only a few meters, so the
proposed method could lead to erroneous predictions. A second limitation is that the current results are obtained from
numerical experiments and not from evaluations of actual observations. Currently, the Lidar system is being modified
to be smaller and lighter in order to suit small experimental aircraft. The onboard Lidar system and real-time airflow-
vector estimation will be validated by flight experiments in 2021; the whole gust-alleviation system, including preview
control, will be demonstrated in 2022. The results of this research will be applied to this flight demonstration.


**Author Contributions**
Ryota Kikuchi, Takashi Misaka, and Shigeru Obayashi designed the experiments. Ryota Kikuchi performed
the experiments, developed the model code, performed the simulations, and prepared the manuscript with
contributions from all co-authors. Hamaki Inokuchi contributed to the analysis and interpretation of data related to
Lidar and assisted in the preparation of the manuscript. All authors approve the final version of the manuscript and
agree to be accountable for all aspects of the work in ensuring that questions related to the accuracy or integrity of any
part of the work are appropriately investigated and resolved.
**Competing Interests**
The authors declare that they have no conflict of interest.

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
