# Peer review of "Real-Time Estimation of Airflow Vector based on Lidar"

_Atmospheric Measurement Techniques, 2020_

## Referee Comment (RC1) · Anonymous Referee #1 · 22 Jul 2020

**Referee Report: Kikuchi et al**

General comments: The paper describes a method that at least partially addresses a fundamental problem of Doppler lidar, that only one component of the 3D wind vector is accessible from a single line-of-sight measurement. They link the work to an important practical requirement related to aircraft safety. The approach seems clearly presented and the results are interesting. The approach of combining lidar returns from different times does have some similarities with a method previously demonstrated by the ZHAW group and it is worth citing the following presentation, slide 51: https://presentations.copernicus.org/EMS2017/EMS2017-322_presentation.pdf

Specific comments:

P2, line 41: "Lidar can detect wind velocity in clear air, but cannot work during precipitation." In fact, it has been well demonstrated that horizontal wind speed can be correctly measured in rain.

P7, line 193: the authors should list the possible physical origins of the random noise sources in their proposed lidar observation method.

P21, line 428: "Flight demonstrations are to be performed in 2021. The results of this research will be applied to this flight demonstration." If possible, it would be useful and interesting if the authors can provide some more detail of their proposed experimental campaign.

Technical corrections:

P3, line 95: "Lidars are assumed to **be** compliant…"

P4, line 121: "the estimation accuracy of the vertical wind velocity is required to be lower than 2.6 ms-1 in the LOS distance of 500 m" – I suggest use of the word "better" rather than "lower".

P11, line 258: "opening size of optical antenna" – specify radius or diameter?

---

## Referee Comment (RC2) · Anonymous Referee #2 · 24 Jul 2020

Within this manuscript, the authors present a novel technique for measuring the headwind/tailwind and vertical velocity ahead of an aircraft for preview control using Doppler wind lidar. The authors present the method and use simulated observations to assess the accuracy of the method, with respect to traditional approaches. The authors demonstrate using these simulations that the proposed technique is more accurate and will meet requirements for preview control. However, the authors do discuss in the conclusions some limitations in detecting small-scale turbulence (i.e., wakes from other aircraft likely cannot be properly detected).

This paper fits well into AMT's scope and will be of interest to its readers, particularly those in the aviation applications community. As such, I suggest that this manuscript be accepted contingent on the following comments are sufficiently addressed to clarify

several aspects of the paper and methodology to readers.

Specific Comments

a) Line 27: Add a reference to this 2014 FAA study or report.

b) Line 28: How many fatal accidents were observed in this time frame?

c) Line 34: Are these turbulence-related accidents results in LOC-I? I don't exactly follow the logic presented here, and if LOC-I accidents are related to CAT (which is the main topic of the paper). Please clarify how statistics in this whole paragraph are related to each other and relevant.

d) Line 38: There are numerical predictions of CAT. Please add a statement and provide any supporting references for if/why these are insufficient for avoiding CAT.

e) Line 42: Rephrase the sentence 'Aerosol particles are received instead of laser beams due to a scattering light effect caused by the rain particles', as it is unclear. Does this mean that the dominant signal comes from raindrops, which are not passive tracers of air motion (whereas aerosols can be safely assumed to be passive tracers due to their small size).

f) Line 48: How far in front of an aircraft can CAT be detected?

g) Line 75: This is not true. A single lidar can measure the vertical velocity if the beam is scanned at different angles (e.g., upwards and downwards using a prism). It is only not possible when the beam is fixed.

h) Line 92: Clarify there and throughout that the horizontal wind (u and v) is not measured, rather only the component that is parallel to the look-direction (i.e., headwind/tailwind component, not crosswind).

i) Sect. 2.1: Please provide more details on the Doppler lidar itself. Does it use heterodyne- or direct-detection? What is its wavelength, PRF, pulse energy, bandwidth, aperture diameter, etc? How many pulses are averaged? What is the data rate?

Also list any other relevant details and provide a written description of the system.

j) Line 120: Where do these requirements for the frequency/accuracy come from?

k) Line 131: Change 'areas' to 'distance', as area connotes a 2-D space. Also change the wording 'area' in Fig.2.

l) Figure 2: The circles in the figure make it seem like the lidars are sweeping out a scan circle (similar to a conical scan). This is not true, suggest removing the circles.

m) Line 156: Why was a first-degree polynomial used and not a higher-order polynomial?

n) Sect. 2.3: It would be helpful if a figure could be added showing one more multiple example spectra (preferably for both a valid and invalid measurement), also showing what these values ($k\_1st$, $k\_2nd$, $k\_ave$) signify.

o) Lines 175-189: It would also be help to add a figure showing how exactly the LSM estimation is used to QC bad measurements. Once invalid measurements are removed following this LSM quality-control process, is the initial LSM fitting done again to obtain a better estimate of the wind? I would think this process could be repeated until there are no more poor LOS estimates going into the fitting.

p) Figure 5: It makes sense to show the vertical velocity with its sign, not just it's magnitude, in this plot, as the sign is expected to change around the vortex.

q) Figure 6: I don't really think this figure is valuable and it could be removed. The only information it contains that is not in the text is the dimensions of the model volume, which could be stated in the text instead.

r) Figure 7: I suggest using a divergent colorbar (i.e., a colorbar that is either white or grey for w=0), which is typically used for vertical velocity. It would be helpful to add the modeled flight path to this figure.

s) Lines: 271-275: These lines would be best in the caption for Figs. 9 and 10, that way

the reader doesn't need to refer back (several pages in the text) to understand what is in Fig. 9 and 10 when examining those figures.

t) Line 275: Clarify what is meant exactly by 'after 10 s or 15 s'?

u) Line 301 and 331, 389: How are the pseudo-routes generated? Are they at random locations in the vortex, or staggered? Are there limits to the heights that they are limited to?

v) Lines 322-325: These lines also would be best in the caption for Figs. 11 and 12.

w) Fig. 12: The colorbar axis on the right plots is not correct. It shows vertical winds of 8 m/s, but the values shown for the NWP output (Fig. 7) did not exceed 2 m/s.

x) Lines 391-392: This text should be moved to the caption to describe what each panel indicates.

y) Line 425: One notable exception to this is wakes from other (larger) aircrafts. These localized but intense vortices pose a safety hazard. The authors should specifically state here that these hazards will not be detected with this technique.

Editorial Corrections

a) Line 87: Change 'smaller than that in the atmosphere' to 'smaller than at lower altitudes'.

b) Line 151: Remove one instance of the word 'method'.

c) Figure 3: The top-middle lidar beam should be labeled T-1, not T-2.

---

## Author Response (AR1)

RESPONSE TO REFEREE #1:

We thank Referee #1 for taking the time to review the manuscript and provide valuable and constructive feedback. The referee's comments are reproduced in bold; excerpts from the revised manuscript are in red type.

**P2, line 41: "Lidar can detect wind velocity in clear air, but cannot work during precipitation." In fact, it has been well demonstrated that horizontal wind speed can be correctly measured in rain**.

We have modified the manuscript as follows:

Emitted laser light is scattered by fine aerosol particles in the atmosphere; the back-scattered light is condensed by telescopes and received by an optical transceiver. Since the wavelength of the received light varies according to the velocity of the aerosol particles due to the Doppler effect, wind speed can be calculated by comparing this wavelength with that of the received light (Inokuchi and Akiyama, 2019). However, when rain is too heavy, the backscattering signal is weakened due to strong attenuation by raindrops and a decrease in aerosols (Wei et. al 2019), making it difficult to measure the wind velocity at a distance.

**P7, line 193: the authors should list the possible physical origins of the random noise sources in their proposed lidar observation method.**

We have modified the manuscript as follows:

The random noise is caused by the reduced intensity of the received light due to the thin aerosol concentration in the sky. A general Lidar signal consists of random noise superimposed on the spectral signal. If the signal intensity is low, random noise may be detected by peak search. (Additional randomness caused by environmental factors and data processing in Lidar is considered here as randomness of the wind-speed values.)

**P21, line 428: "Flight demonstrations are to be performed in 2021. The results of this research will be applied to this flight demonstration." If possible, it would be useful and interesting if the authors can provide some more detail of their proposed experimental campaign.**

We have modified the manuscript as follows:

Currently, the Lidar system is being modified to be smaller and lighter in order to suit small experimental aircraft. The onboard Lidar system and real-time airflow-vector estimation will be validated by flight experiments in 2021; the whole gust-alleviation system, including preview control, will be demonstrated in 2022. The results of this research will be applied to this flight demonstration.

**P3, line 95: "Lidars are assumed to be compliant..."**

We have modified the manuscript as follows:

The Lidars are assumed to be compliant with the specifications for preview control currently under development by the JAXA.

**P4, line 121: "the estimation accuracy of the vertical wind velocity is required to be lower than 2.6 ms-1 in the LOS distance of 500 m" – I suggest use of the word "better" rather than "lower".**

We have modified the manuscript as follows:

... the estimation accuracy of the vertical wind velocity must be better than 2.6 m s$^{-1}$ ...

**P11, line 258: "opening size of optical antenna" – specify radius or diameter?**

We have modified the manuscript as follows:

The Lidar sensor is shown in Fig. 2; its specifications are given in Table 1 (Inokuchi and Akiyama 2019). Laser pulses generated by an optical transceiver are amplified by optical amplifiers (Sakimura et. al. 2013) incorporated into an optical antenna and radiated into the atmosphere from optical telescopes. The heat generated by the optical amplifiers is dissipated by a chiller unit using water as a coolant. The optical antenna is equipped with a 150 mm large-aperture telescope for long range observations and a 50 mm small-aperture telescope for vector conversion of short-range observations.

**The approach of combining lidar returns from different times does have some similarities with a method previously demonstrated by the ZHAW group and it is worth citing the following presentation, slide 51:**

**https://presentations.copernicus.org/EMS2017/EMS2017-322_presentation.pdf**

We have modified the manuscript as follows:

Neininger, B.: Trends in airborne atmospheric observations, European Meteorological Society Annual Meeting 2017,  14, EMS2017-322, 2017

RESPONSE TO REFEREE #2:

We thank Referee #2 for taking the time to review the manuscript and provide valuable and constructive feedback. The referee's comments are reproduced in bold; excerpts from the revised manuscript are in red type.

**Specific Comments**

**a) Line 27: Add a reference to this 2014 FAA study or report.**

We have modified the manuscript as follows:

For both fatal and non-fatal aircraft accidents, the impact of atmospheric turbulence can be significant. The Japan Transport Safety Board has stated that accidents caused by turbulence accounted for 48% of non-fatal aircraft accidents in Japan involving commercial airplanes from 2003-2012. An increase in the rate of accidents related to turbulence was reported by the Federal Aviation Administration in 2006, Kim and Chun in 2011, and Williams in 2017.

**b) Line 28: How many fatal accidents were observed in this time frame?**

We have modified the manuscript as follows:

Statistics reported by Boeing (2018) show that 322 non-fatal and 51 fatal accidents occurred worldwide in commercial jet flights from 2009 through 2018.

**c) Line 34: Are these turbulence-related accidents results in LOC-I? I don't exactly**

**follow the logic presented here, and if LOC-I accidents are related to CAT (which is**

**the main topic of the paper). Please clarify how statistics in this whole paragraph are**

**related to each other and relevant.**

Aircraft accidents are often caused by a combination of factors that are difficult to completely identify. LOC-I can be caused by human factors, environmental factors, system factors, or a combination of these. First, the statistics show that wind phenomena (wind shear and atmospheric turbulence) are significant factors in LOC-I. Second, the statistics of aircraft accidents in Japan show the atmospheric turbulence is the largest factor in non-fatal accidents in Japan. Although non-fatal accidents (mentioned in the Japanese statistics) and fatal ones (mentioned in the LOC-I statistics) are quite different in their impact, turbulence affects both. Non-fatal accidents include events such as broken bones in passengers not wearing their seat belts and burns from spilling hot coffee while serving a cabin attendant.

We have modified the manuscript as follows:

Atmospheric turbulence poses a potential risk to aircraft operation. Statistics reported by Boeing (2018) show that 322 non-fatal and 51 fatal accidents occurred worldwide in commercial jet flights from 2009 through 2018. Of the fatal accidents, the largest proportion (25.5%) were due to Loss of Control-In Flight (LOC-I). The International Air Transportation Association (2016) shows that LOC-I frequently occurs when the aircraft speed is well below the stall speed; in conjunction with weather conditions, low speed is the most common factor in LOC-I accidents. Forty-two percent of LOC-I accidents occurred under degraded meteorological conditions affecting aircraft speed, in particular strong wind shear and atmospheric turbulence.

For both fatal and non-fatal aircraft accidents, the impact of atmospheric turbulence can be significant. The Japan Transport Safety Board has stated that accidents caused by turbulence accounted for 48% of non-fatal aircraft accidents in Japan involving commercial airplanes from 2003-2012. An increase in the rate of accidents related to turbulence was reported by the Federal Aviation Administration in 2006, Kim and Chun in 2011, and Williams in 2017.

**d) Line 38: There are numerical predictions of CAT. Please add a statement and provide any supporting references for if/why these are insufficient for avoiding CAT.**

We have modified the manuscript as follows:

Numerical weather prediction (NWP), which is an essential tool for aircraft operation, can forecast weather conditions for days and even weeks in advance and output broader-area weather information than can Radar or Lidar. However, NWP cannot explicitly resolve disturbances as small as most turbulence, leading to a very large predictive uncertainty (Sharman et al. 2006, Kim et al. 2011). Therefore, some researchers have developed an alternative approach that predicts turbulence potential by calculating turbulence indicators from NWP results; for example, Sharman et al. (2006) have developed an approach called graphical turbulence guidance (GTG) that combines such indicators. The turbulence potential can also be used to determine operational flight routes (Kim et al. 2015), but it has a large spatio-temporal gap on the scale of aircraft motion because it is based on NWP results such as the meso-scale model. It thus provides insufficient information to implement turbulence avoidance on aircraft in flight.

**e) Line 42: Rephrase the sentence 'Aerosol particles are received instead of laser beams due to a scattering light effect caused by the rain particles', as it is unclear. Does this mean that the dominant signal comes from raindrops, which are not passive tracers of air motion (whereas aerosols can be safely assumed to be passive tracers due to their small size).**

We have modified the manuscript as follows:

Emitted laser light is scattered by fine aerosol particles in the atmosphere; the back-scattered light is condensed by telescopes and received by an optical transceiver. Since the wavelength of the received light varies according to the velocity of the aerosol particles due to the Doppler effect, wind speed can be calculated by comparing this wavelength with that of the received light (Inokuchi and Akiyama, 2019). However, when rain is too heavy, the backscattering signal is weakened due to strong attenuation by raindrops and a decrease in aerosols (Wei et. al 2019), making it difficult to measure the wind velocity at a distance.

**f) Line 48: How far in front of an aircraft can CAT be detected?**

We have modified the manuscript as follows:

Inokuchi et al. (2012) have shown observationally that airborne Doppler Lidar can detect CAT in front of an aircraft in flight at altitudes of 3,200 m; the Lidar information can be detected 30 seconds before the turbulence affects the aircraft. The aircraft's flight speed in the test was 320 kt (160 m/s), so it detected CAT from a distance of about 4.8 km.

**g) Line 75: This is not true. A single lidar can measure the vertical velocity if the beam**

**is scanned at different angles (e.g., upwards and downwards using a prism). It is only**

**not possible when the beam is fixed.**

We have modified the manuscript as follows:

However, a fixed single Doppler Lidar system can only detect the line-of-sight (LOS) wind, providing a one-dimensional piece of information; the vertical wind velocity in front of the aircraft cannot be measured by such a system (Hamada, 2019). It is necessary to perform the Lidar measurements in two directions, upward and downward, to obtain the vertical wind velocity (Neininger, 2017).

**h) Line 92: Clarify there and throughout that the horizontal wind (u and v) is not measured, rather only the component that is parallel to the look-direction (i.e., headwind/tailwind component, not crosswind).**

We have modified the manuscript as follows:

In this study, "horizontal wind" means any headwind/tailwind component that does not include the crosswind component.

**i) Sect. 2.1: Please provide more details on the Doppler lidar itself. Does it use heterodyne- or direct-detection? What is its wavelength, PRF, pulse energy, band-width, aperture diameter, etc? How many pulses are averaged? What is the data rate?Also list any other relevant details and provide a written description of the system.**

We have modified the manuscript as follows:

The Lidar sensor is shown in Fig. 2; its specifications are given in Table 1 (Inokuchi and Akiyama 2019). Laser pulses generated by an optical transceiver are amplified by optical amplifiers (Sakimura et. al. 2013) incorporated into an optical antenna and radiated into the atmosphere from optical telescopes. The heat generated by the optical amplifiers is dissipated by a water-cooled chiller unit. The optical antenna is equipped with a 150 mm large-aperture telescope for long range observations and a 50 mm small-aperture telescope for vector conversion of short-range observations.

**j) Line 120: Where do these requirements for the frequency/accuracy come from?**

We have modified the manuscript as follows:

The control requirements are the conditions necessary to halve the peak variation in acceleration by control. This value has been specified using control simulations (Hamada, 2019), and Monte Carlo simulations have also been performed.

**k) Line 131: Change 'areas' to 'distance', as area connotes a 2-D space. Also change**

**the wording 'area' in Fig.2.**

We modified the Figure 2 (Figure 3 in revised paper) according to the referee's comment. See answer to (l) below.

**l) Figure 2: The circles in the figure make it seem like the lidars are sweeping out a**

**scan circle (similar to a conical scan). This is not true, suggest removing the circles.**

We have modified Fig. 2 (now Fig. 3) as follows:

[Figure]

**Fig. 3      Distance to wind-field region between the Lidars for two line-of-sight (LOS) distances**

**m) Line 156: Why was a first-degree polynomial used and not a higher-order polynomial?**

We have modified the manuscript as follows:

Depending on the number of past LOS wind data used, the order of the polynomial expression used in the extrapolation varies. The aerosol concentration in the upper sky is low, suggesting that there is considerable missing data and noise. A sufficient number of past LOS wind data may not be available to estimate a high-order polynomial expression, and this could affect the robustness of the control. For this reason, a first-degree polynomial expression is adopted in this study and used in the least-squares method (LSM) to extrapolate the wind-field values according at the horizontal line. The airflow vector is calculated by Eq. (1) using the extrapolated LOS wind.

**n) S**ect. 2.3: It would be helpful if a figure could be added showing one more multiple example spectra (preferably for both a valid and invalid measurement), also showing what these values (k_1st, k_2nd, k_ave) signify.

We have modified the manuscript as follows:

Figure 5 shows a conceptual explanation of the variables of simplified Kalman gain in the cases of correct measurement and of an error peak. In this study, the filtering algorithm is carried out first when the observation data is obtained:

$$K = \begin{cases} 1 & |k_{1st} - k_{2nd}| = 1 \ and \ |k_{1st} - k_{ave}| < k_{dif} \\ 0 & Otherwise \end{cases}$$

(4)

[Figure]

**Fig. 5**    Conceptual explanation of the variables of simplified Kalman gain.

(a) Correct measurement case of *K=1*.  (b) Case with the error peak of *K=0*

**o) Lines 175-189: It would also be help to add a figure showing how exactly the LSM**

**estimation is used to QC bad measurements. Once invalid measurements are removed**

**following this LSM quality-control process, is the initial LSM fitting done again to obtain**

**a better estimate of the wind? I would think this process could be repeated until there**

**are no more poor LOS estimates going into the fitting.**

We have added a figure, as the referee suggested, and modified the manuscript as follows:

Figure 6 explains the concept behind Tuckey's biweight methodology as applied to Lidar. The concept of a robust LSM is validated by analyzing the difference between the observed LOS wind values and those estimated by the polynomial expression used in LSM. In the 1st step, the LOS wind is estimated by using the general LSM (Eq. (2)). In the 2nd step, the difference $d_j{}^T$ between the observed LOS wind value and that estimated from the polynomial expression is found:

$$d_j{}^T = W_j{}^T - (a_j z + b_j).$$
(5)

A permissible difference range $L$ is defined and weights $w_j{}^T (d_j{}^T)$ are calculated depending on where $d_j{}^T$ falls in the distance range:

$$w_j{}^T (d_j{}^T) = 0 \ (d_j{}^T < -L)$$

$$w_j{}^T (d_j{}^T) = \left(1 - \left(\frac{d_j{}^T}{w_j{}^T}\right)^2\right)^2 \ (-L \le d_j{}^T \le L) .$$
(6)

$$w_j{}^T (d_j{}^T) = 0 \ (d_j{}^T > L)$$

Weights are assigned to each LOS wind velocity value. In the 3rd step, a new first-degree polynomial expression for the LSM with the weighted data is estimated as follows.

$$a_j{}' = \frac{\sum_{i=T-(N-1)}^{T} w_j{}^i \sum_{i=T-(N-1)}^{T} w_j{}^i z^i W_j{}^i - \sum_{i=T-(N-1)}^{T} w_j{}^i z^i \sum_{i=T-(N-1)}^{T} w_j{}^i W_j{}^i}{\sum_{i=T-(N-1)}^{T} w_j{}^i \sum_{i=T-(N-1)}^{T} w_j{}^i (z^i)^2 - \left(\sum_{i=T-(N-1)}^{T} w_j{}^i z^i\right)^2}$$

$$b_j{}' = \frac{\sum_{i=T-(N-1)}^{T} w_j{}^i z^i \sum_{i=T-(N-1)}^{T} w_j{}^i W_j{}^i - \sum_{i=T-(N-1)}^{T} w_j{}^i z^i W_j{}^i \sum_{i=T-(N-1)}^{T} w_j{}^i z}{\sum_{i=T-(N-1)}^{T} w_j{}^i \sum_{i=T-(N-1)}^{T} w_j{}^i (z^i)^2 - \left(\sum_{i=T-(N-1)}^{T} w_j{}^i z^i\right)^2}$$
(7)

This process is repeated until the weight of the error value decreases and converges.

[Figure]

**Fig. 6 Conceptual explanation of Tuckey's biweight methodology applied to line-of-sight (LOS) wind at various distances. First step: simple least-squares fit. Second step: observations are compared with the estimate. The data are weighted, and extreme outliers are excluded, using Eq. (6). Third step: Least-squares fit of the weighted data.**

**p) Figure 5: It makes sense to show the vertical velocity with its sign, not just it's**

**magnitude, in this plot, as the sign is expected to change around the vortex.**

We modified the Figure 5 (Figure 8 in revised paper) according to the referee's comment. As the referee comments, it is important to specify the direction of rotation of a vortex because the vertical wind speed changes from positive to negative around the vortex. Therefore, the direction of rotation of the vortex is now indicated by arrows in the revised figure.

[Figure]

**q) Figure 6: I don't really think this figure is valuable and it could be removed. The only information it contains that is not in the text is the dimensions of the model volume, which could be stated in the text instead.**

Figure 6 has been removed, in agreement with the referee's comment.

**r) Figure 7: I suggest using a divergent colorbar (i.e., a colorbar that is either white or grey for w=0), which is typically used for vertical velocity. It would be helpful to add the modeled flight path to this figure.**

We modified Figure7 (Figure 9 in revised paper) according to the referee's comment. Because of the difference in the maximum and minimum vertical winds at the top and bottom and the difficulty of setting the divergent colorbar center to 0 m/s, the colorbar was left as it was. However, an example of an aircraft flight path is shown.

[Figure]

**Fig. 9**  Vertical wind velocity distribution map generated by JMA-NHM

**s) Lines: 271-275: These lines would be best in the caption for Figs. 9 and 10, that way**

**the reader doesn't need to refer back (several pages in the text) to understand what is**

**in Fig. 9 and 10 when examining those figures.**

We have modified the manuscript as follows (note that Fig. 9 is now Fig. 11):

**Fig. 11**  Distributions of the horizontal and vertical wind components estimated by the simple vector conversion method vs. the proposed method (at time 10 s). Upper figures: ideal vortex model; middle figures: simple vector conversion method; lower figures: proposed method with five-past LOS wind datasets. Left figures: horizontal wind values; right figures: vertical wind values and a similar for Fig. 12 (at time 15 sec).

**t) Line 275: Clarify what is meant exactly by 'after 10 s or 15 s'?**

We have modified the manuscript as follows:

Figures 11 and 12 show the results of starting the flight from the edge of the computational space, 10 and 15 seconds later and 15 seconds later. Thus, they represent the time before and during the aircraft's close approach to the vortex core.

**u) Line 301 and 331, 389: How are the pseudo-routes generated? Are they at random**

**locations in the vortex, or staggered? Are there limits to the heights that they are limited**

**to?**

We have modified the manuscript as follows:

Line 353: Next, the statistical estimation performance is evaluated by using 100 pseudo-routes that are randomly generated 750 m up and down from the center of the vortex core.

Line 386: Next, the statistical estimation performance is evaluated using 100 pseudo-routes that are randomly generated between 2 km and 10 km altitude.

Line 418: Next, the statistical estimation performance [in this case, different from that in Line 353] is evaluated by using 100 pseudo-routes that are randomly generated 750 m up and down from the center of the vortex core.

**v) Lines 322-325: These lines also would be best in the caption for Figs. 11 and 12.**

We modified the sentences and added the explanation to the caption according to the referee's comment.

**w) Fig. 12: The colorbar axis on the right plots is not correct. It shows vertical winds of**

**8 m/s, but the values shown for the NWP output (Fig. 7) did not exceed 2 m/s.**

The right colorbar of Fig. 12 (now called Fig. 14) was indeed incorrect and has been modified:

[Figure]

**x) Lines 391-392: This text should be moved to the caption to describe what each**

**panel indicates.**

We modified the sentences and added the explanation to the caption according to the referee's comment.

**y) Line 425: One notable exception to this is wakes from other (larger) aircrafts. These**

**localized but intense vortices pose a safety hazard. The authors should specifically**

**state here that these hazards will not be detected with this technique.**

We have modified the manuscript as follows:

An exception to this is aircraft-generated wake turbulence, which still poses a safety risk. The radius of the actively fluctuating wake-turbulence core is only a few meters, so the proposed method could lead to erroneous predictions.

**Editorial Corrections**

**a) Line 87: Change 'smaller than that in the atmosphere' to 'smaller than at lower**

**altitudes'.**

We have modified the manuscript as follows:

In addition, actual Lidar observations involve errors, noise, and loss of data, with  negative effects on aircraft control, as reported by Misaka et al. (2015); these problems are worse at higher altitudes, where the aerosol density is smaller than it is at lower ones.

**b) Line 151: Remove one instance of the word 'method'.**

We have modified the sentence according to the referee's comment.

**c) Figure 3: The top-middle lidar beam should be labeled T-1, not T-2.**

We have modified the manuscript as follows:

[revised manuscript text omitted]